# A Mixed-Curvature based Pre-training Paradigm for Multi-Task Vehicle Routing Solver

**Suyu Liu** [1]  **Zhiguang Cao** [2]  **Shanshan Feng** [3]  **Yew-Soon Ong** [1 4]

## Abstract

Solving various types of vehicle routing problems (VRPs) using a unified neural solver has garnered significant attentions in recent years. Despite their effectiveness, existing neural multi-task solvers often fail to account for the geometric structures inherent in different tasks, which may result in suboptimal performance. To address this limitation, we propose a curvature-aware pre-training framework. Specifically, we leverage mixed-curvature spaces during the feature fusion stage, encouraging the model to capture the underlying geometric properties of each instance. Through extensive experiments, we evaluate the proposed pre-training strategy on existing neural multi-task solvers across a variety of testing scenarios. The results demonstrate that the curvature-aware pre-training approach not only enhances the generalization capabilities of existing neural VRP solvers on synthetic datasets but also improves solution quality on real-world benchmarks.

## 1. Introduction

Vehicle routing problems (VRPs) owing to its broad applicability among various domains such as transportation service (Ge et al., 2019; Zhou et al., 2023a) and trajectory planning (Dantzig & Ramser, 1959; Min, 1989), have garnered great attentions in recent years. However, because of its NP-Hard complexity, obtaining optimal solutions within a reasonable time is almost infeasible. Regarding to this, several heuristic solvers have been proposed, such as Lin-

Kernighan-Helsgaun (LKH) (Lin & Kernighan, 1973; Helsgaun, 2017), Hybrid Genetic Search (HGS) (Vidal, 2022) and OR-Tools (Perron & Didier, 2024). Despite the fact that these solvers have achieved remarkable performances, their reliance on hand-crafted rules and specialized domain knowledge may severely limit their abilities to generalize to more general problem types, especially the emerging VRP variants. Additionally, these solvers may become computationally prohibitive when applied on large-scale instances.

On the other hand, deep learning based neural solvers such as (Kool et al., 2019; Kwon et al., 2020; Zhou et al., 2023b;a; Goh et al., 2024; Zhang et al., 2025) require minimal hand-crafted rules and offer significantly much faster inference speeds. Typically, following the architecture of POMO (Kwon et al., 2020), these neural solvers often contain self-attention modules and utilize reinforcement learning as their optimization algorithms. Besides, hard masks are integrated into the attention mechanism to eliminate infeasible actions. Despite their efficiency and flexibility, these solvers primarily address relatively simple VRPs, and extending current frameworks to handle more complex problem types remains as an under-explored area.

Recently, there has been a growing trend towards building multi-task foundation models for solving various types of VRPs (Liu et al., 2024; Zhou et al., 2024; Berto et al., 2024; Huang et al., 2025). Although these models have demonstrated promising results, they largely overlook the geometric structures that widely exist in different tasks. As shown in Figure 1, VRP instances, despite being defined in Euclidean coordinates, exhibit node-level curvature distributions that cannot be faithfully captured in flat spaces: the Ollivier-Ricci curvature (defined in Eq. (15), Appendix.1) which effectively quantifies geometric structures on discrete spaces like graphs or networks, reveals that almost every node carries either negative or positive curvature, indicating that the underlying data contains structures in a mixed-curvature space rather than a purely Euclidean one. Specifically, we observe that nodes are frequently situated in regions of either positive (contractive) or negative (expansive) curvature spaces, which correlate with delivery patterns such as customer clustering or route divergence in the delivery map (Figure 5, Appendix.1). All of these factors

[1]College of Computing and Data Science, Nanyang Technological University, Singapore [2]School of Computing and Information Systems, Singapore Management University, Singapore [3]Centre for Frontier AI Research, The Agency for Science, Technology and Research, Singapore [4]Centre for Frontier AI Research, Institute of High Performance Computing, Agency for Science, Technology and Research, Singapore. Correspondence to: Zhiguang Cao <zgcao@smu.edu.sg>.

*Proceedings of the 42$^{nd}$ International Conference on Machine Learning*, Vancouver, Canada. PMLR 267, 2025. Copyright 2025 by the author(s).

are critical to decision makings in solving vehicle routing problems. However, the embedding and feature transformation spaces in current neural solvers are still confined to Euclidean geometric spaces where each point (or node) is treated uniformly, severely limiting their abilities to adapt to such heterogeneous geometries (Nickel & Kiela, 2017; Ganea et al., 2018; Liu et al., 2019; Chami et al., 2019; Desai et al., 2023). Indeed, prior work (Sala et al., 2018) has shown that Euclidean spaces, regardless of dimensionality, struggle to represent complex structures such as trees without incurring significant distortions. Fortunately, some deep learning methods on Riemannian manifolds (Nickel & Kiela, 2017; Ganea et al., 2018; Gu et al., 2018) have provided an alternative way to avoid these potential pitfalls.

In this work, we propose the first pre-training strategy that trains multi-task foundation models within a mixed-curvature geometric space to solve various types of VRPs, which empowers the neural solvers with the ability to capture nuanced geometric information from inputs in a curvature-sensitive manner. Specifically, we partition the feature space of each encoder layer into multiple subspaces, each mapped to a geometric space of a specific curvature. Features from the previous layer are projected into these distinct curvature spaces and subsequently merged in the output stage. By leveraging the unique properties of non-Euclidean spaces such as hyperbolic (negative curvature) and hyperspherical (positive curvature) geometries, our approach allows the model to effectively capture complex geometric patterns from problem instances, offering a novel way to enhance performance across a wide range of VRPs. Accordingly, our contributions are summarized as follows:

- We investigate the multi-task VRP problem from a novel perspective by introducing mixed-curvature geometric spaces, motivated by the diverse curvatures of nodes. To our knowledge, this is the first work to explore a curvature-aware neural solver for VRPs.

- We propose a novel and practical pre-training paradigm that integrates spaces of varying curvatures, enabling the model to explore inherent geometric structures from the inputs for solving VRPs.

- Through extensive experiments, we demonstrate that our proposed approach not only achieves remarkable improvements across various types of VRPs but also shows its strong adaptability to different architectures of multi-task solvers. In addition, results on real-world benchmarks further validate its effectiveness.

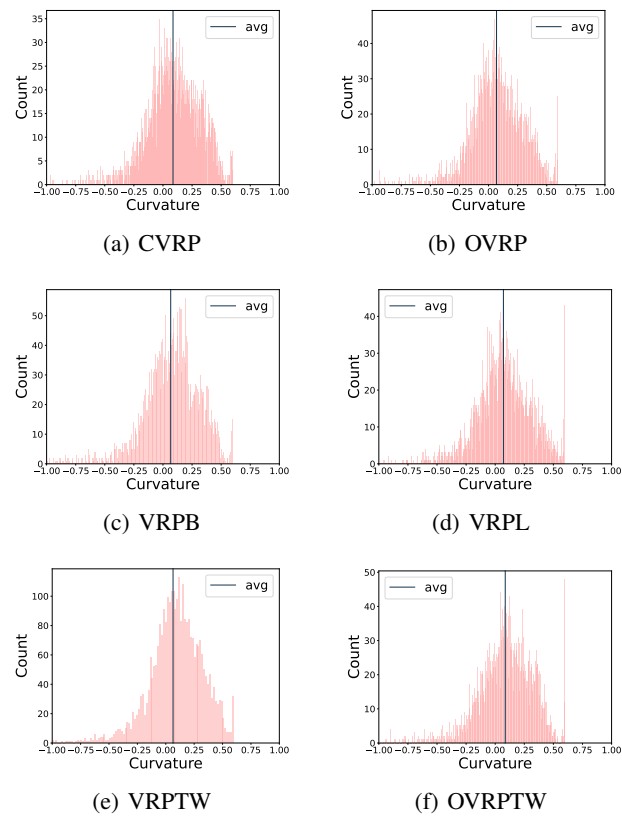

(a) CVRP        (b) OVRP

(c) VRPB        (d) VRPL

(e) VRPTW        (f) OVRPTW

*Figure 1.* Histograms of curvatures for each node across 6 VRP tasks. We utilize 1,000 instances for every task, each containing 50 nodes, to visualize curvature distributions. The x-axis represents curvature values, while the y-axis denotes the count of each value. The *avg* line indicates the average curvature across all nodes. We employ Ollivier-Ricci curvature (Ollivier, 2009) which is well-suited for measuring curvatures in discrete structures like graphs. For further details on this curvature, please refer to Appendix.1. It is demonstrated that almost every node in the dataset has either negative or positive curvature and the average curvature suggests that each task in the Euclidean space contains non-Euclidean geometry information, motivating the use of a mixed-curvature space. Visualizations of curvatures for other VRP tasks are provided in Figure 4, Appendix.1. Better viewed in color.

## 2. Related Work

**VRP Solvers**

Existing solvers for VRPs can be broadly classified into three categories: 1) Traditional Solvers: This category includes established methods such as Lin-Kernighan-Helsgaun (LKH) algorithm (Lin & Kernighan, 1973), Hybrid Genetic Search (HGS) (Vidal, 2022), and OR-Tools (Perron & Didier, 2024). These solvers leverage heuristic search algorithms and rely heavily on expert knowledge, which may limit their adaptabilities to new problem settings. 2) Neural Solvers: Building on early works like (Vinyals

et al., 2015), these methods employ deep learning to iteratively construct solutions. The introduction of self-attention (Vaswani et al., 2017) to VRPs (Kool et al., 2019; Kwon et al., 2020) has significantly improved solution quality. Subsequent progress, such as (Kim et al., 2022; Zhou et al., 2023b), focuses on training with varied data to enhance generalization to unseen scenarios. Recent developments include scaling to larger problem instances (Luo et al., 2023; Pan et al., 2023; Ye et al., 2024; Cheng et al., 2023) and exploring non-autoregressive decoding (Sun & Yang, 2023). However, these methods often rely on additional heuristic searches for particularly challenging or large-scale instances, which may limit the efficiency. 3) Hybrid Solvers: These approaches combine the strength of neural approaches with traditional heuristics to overcome their own limitations. Examples include the adaptation of neural methods for candidate set generation (Xin et al., 2021), enhancing the flexibility and efficiency of classic architectures. Hybrid solvers like (Hottung & Tierney, 2020; Hottung et al., 2021; Xin et al., 2021; Chalumeau et al., 2023; Ma et al., 2024; Chen et al., 2024) have demonstrated considerable success. However, these methods often require task-specific training, which hinders their ability to generalize across different VRPs.

Recent efforts (Liu et al., 2024; Zhou et al., 2024; Berto et al., 2024) have begun focusing on cross-task learning to address the generalization gaps observed in earlier approaches. For instance, (Liu et al., 2024) introduces attribute composition to handle a wide range of VRP variants, while (Zhou et al., 2024) employs a mixture-of-experts (MoE) framework to balance performance and computational efficiency. Our method diverges from these by leveraging mixed-curvature spaces to process input features, enabling more effective capture of intricate geometric structures and providing a generalizable solution across diverse VRP tasks.

**Deep Learning in Non-Euclidean Space**

Unlike the Euclidean setting, which assumes data points lie in flat and homogeneous spaces, non-Euclidean geometry models the underlying space by curved Riemannian manifolds. The family of curved Riemannian manifolds can be broadly categorized into two types: hyperbolic surface (characterized by negative curvature) and hyperspherical surface (characterized by positive curvature). In details, hyperbolic geometry can be expressed through five isometric models, including Poincaré ball model (Nickel & Kiela, 2017; Ganea et al., 2018), Lorentz model(hyperboloid) (Chen et al., 2021; Bdeir et al., 2024; Nickel & Kiela, 2018), Poincaré half space model (Stahl, 1993), Klein model (Bi et al., 2015) and hemisphere model (Cannon et al., 1997). Thanks to their non-uniform distance metric, hyperbolic surfaces are particularly well-suited for extracting hierarchical and relational structures from data and this has led to their wide applications in vision (Khrulkov et al., 2020; Atigh et al.,

2022; Moreira et al., 2024), language (Dai et al., 2021; Fan et al., 2024; Qu et al., 2024), audio (Hong et al., 2023) and data mining (Chami et al., 2019; Liu et al., 2019; Sun et al., 2021; Choudhary et al., 2024). On the other hand, hyperspherical surfaces constrain data representations within a unit hypersphere. This property helps model achieve lower variances and better generalization abilities across a wide range of applications, including image classification (Liu et al., 2017b;a), adversarial attack (Pang et al., 2020) and generative modeling (Qiu et al., 2023). Other works like (Gu et al., 2018; Wang et al., 2021; Sun et al., 2022; Cho et al., 2023; Wang et al., 2024; Fu et al., 2025) have explored a mixed-curvature environment where models process features across spaces with varying curvatures, which leverages the strengths of both hyperbolic and hyperspherical geometries into learning process. In contrast to all these works, ours focuses on learning diverse data representations for variants of VRPs, aiming to enhance the cross-task generalization ability.

## 3. Preliminaries

We introduce essential definitions related to mixed-curvature spaces and key concepts in VRPs. For a broader overview of geometric deep learning, we refer interested readers to the surveys (Peng et al., 2021; Mettes et al., 2024).

### 3.1. Basics of Riemannian Manifolds

A *Riemannian manifold* $\mathcal{M}$ is a smooth structure equipped with a metric $g_{\mathbf{x}}$. This metric is a smoothly varying positive-definite inner product defined on the tangent space $T_{\mathbf{x}}\mathcal{M}$ of point $\mathbf{x} \in \mathcal{M}$. Such kind of structures generalizes the concepts like distance and angle from Euclidean space to more complex geometric spaces. To navigate between the manifold and its tangent space more conveniently, the following two important mappings are often used:

$$Exp_{\mathbf{x}}^{\kappa} : T_{\mathbf{x}}\mathcal{M} \to \mathcal{M}, \quad Log_{\mathbf{x}}^{\kappa} : \mathcal{M} \to T_{\mathbf{x}}\mathcal{M}. \quad (1)$$

The *exponential map*, denoted by $Exp_{\mathbf{x}}^{\kappa}$, transfers vectors from tangent space $T_{\mathbf{x}}\mathcal{M}$ back to manifold $\mathcal{M}$ of curvature $\kappa$. The *logarithmic map*, denoted by $Log_{\mathbf{x}}^{\kappa}$, transfers vectors from manifold $\mathcal{M}$ of curvatre $\kappa$ to tangent space $T_{\mathbf{x}}\mathcal{M}$. Due to page limit, we put their mathematical expressions under hyperbolic and hyperspherical settings in Eqs. (17), (18), (19), (20), Appendix.1.

### 3.2. Hyperbolic and Hyperspherical Spaces

Our framework is built upon a mixed-curvature space that integrates properties of multiple geometric spaces. Below, we provide a brief overview of the two geometric spaces that are employed in this work: the hyperbolic spaces and hyperspherical spaces.

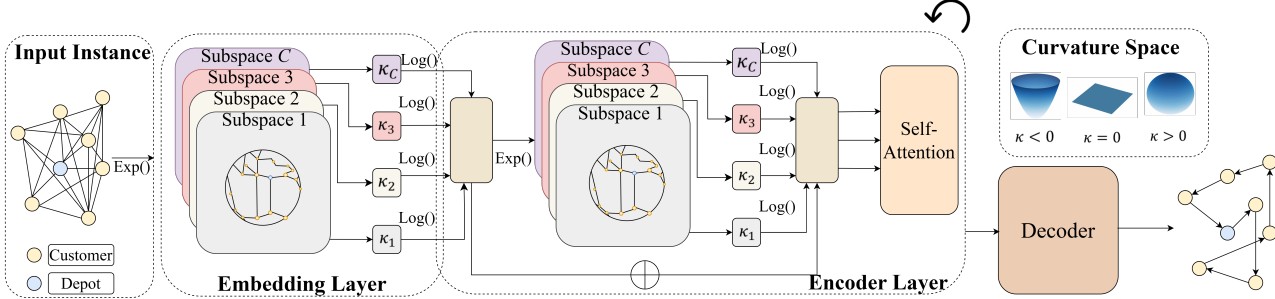

*Figure 2.* The framework of the proposed module. We consider three geometric spaces with negative (hyperbolic), zero (Euclidean) and positive (hyperspherical) curvatures, respectively. For each feature transformation operation, we split original feature space into $C$ smaller subspaces, each with their own learnable curvatures ($\kappa_1, \ldots, \kappa_C$). Operations like $Exp$ and $Log$ are frequently used to navigate vectors between manifold and tangent space. In the encoder layer, an extra *Mix-up* method is utilized to make information transmission smoother from shallow layer to deeper ones.

**Hyperbolic Spaces.** In our work, we adopt the Poincaré ball for modeling hyperbolic geometric information as proposed in (Ganea et al., 2018). Hyperbolic space is characterized by a negative curvature $\kappa < 0$ and its domain is defined as:

$$\mathbb{H}(\kappa) = \{\mathbf{x} \in \mathbb{R}^d | -\kappa \cdot ||\mathbf{x}||_2^2 < 1\}, \quad (2)$$

where $||\mathbf{x}||_2$ is the regular $L_2$ distance. The associated conformal factor is given by $\lambda_{\mathbf{x}}(\cdot, \cdot) = \frac{2}{1+\kappa||\mathbf{x}||_2^2}$. Distance and arithmetic operations are derived in (Ganea et al., 2018). For instance, the addition operation, denoted by $\oplus_\kappa$ takes the following form:

$$\mathbf{x} \oplus_\kappa \mathbf{y} = \frac{(1 - 2\kappa\langle\mathbf{x}, \mathbf{y}\rangle - \kappa\|\mathbf{y}\|_2^2)\mathbf{x} + (1 + \kappa\|\mathbf{x}\|_2^2)\mathbf{y}}{1 - 2\kappa\langle\mathbf{x}, \mathbf{y}\rangle + \kappa^2\|\mathbf{x}\|_2^2\|\mathbf{y}\|_2^2}. \quad (3)$$

Building on this, the distance between two points $\mathbf{x}$ and $\mathbf{y}$ in hyperbolic space can be calculated in the following format:

$$d_\kappa(\mathbf{x}, \mathbf{y}) = \left(\frac{2}{\sqrt{-\kappa}}\right)\tanh^{-1}\left(\sqrt{-\kappa}\| - \mathbf{x} \oplus_\kappa \mathbf{y}\|_2\right). \quad (4)$$

**Hyperspherical Spaces.** The hypersphere, also referred to as a spherical space, is characterized by a positive curvature $\kappa > 0$:

$$\mathbb{S}(\kappa) = \{\mathbf{x} \in \mathbb{R}^d | \kappa \cdot ||\mathbf{x}||_2^2 = 1\}, \quad (5)$$

and the distance between two points $\mathbf{x}$ and $\mathbf{y}$ on the sphere is given as:

$$d_\kappa(\mathbf{x}, \mathbf{y}) = \frac{1}{\sqrt{\kappa}}\cos^{-1}\left(\kappa \cdot \langle x, y\rangle\right), \quad (6)$$

where $\langle\cdot, \cdot\rangle$ is the regular vector inner product.

*Remarks.* In hyperbolic spaces, as $|\kappa|$ increases, the conformal factor $\lambda_{\mathbf{x}}$ decreases, leading to greater distances between points. This causes points on the manifold to spread out in a more noticeable way. Conversely, in hyperspherical spaces, increasing $|\kappa|$ results in shorter distances, drawing points closer together. As $|\kappa|$ approaches zero, both geometries degenerate into the standard Euclidean space.

### 3.3. Product Manifold Spaces

Product manifold space consists of multiple manifolds with different curvatures. It is defined by Cartesian product:

$$\mathcal{M} = \mathcal{M}_1 \times \mathcal{M}_2 \times \cdots \times \mathcal{M}_C, \quad (7)$$

which is equipped with curvature $\overline{\kappa} = (\kappa_1, \cdots, \kappa_C)$. Each point in $\mathcal{M}$ has the form $\overline{\mathbf{x}} = (\mathbf{x}_1, \cdots, \mathbf{x}_C)$. In this case, *exponential map* and *logarithmic map* take following forms:

$$Exp_{\overline{\mathbf{x}}}^{\overline{\kappa}}(\cdot) = (Exp_{\mathbf{x}_1}^{\kappa_1}(\cdot), \cdots, Exp_{\mathbf{x}_C}^{\kappa_C}(\cdot)),$$
$$Log_{\overline{\mathbf{x}}}^{\overline{\kappa}}(\cdot) = (Log_{\mathbf{x}_1}^{\kappa_1}(\cdot), \cdots, Log_{\mathbf{x}_C}^{\kappa_C}(\cdot)). \quad (8)$$

### 3.4. Basics of VRPs

The input of VRP instance (*e.g.* CVRP) is a fully connected, undirected graph $G = (V, E)$, where $V = \{v_0, \ldots, v_n\}$ denotes the set of $n + 1$ nodes including the depot $v_0$ and $n$ customer nodes. The set $E = \{e_{ij}, i, j = 0, \ldots, n\}$ denotes the set of edges and each edge has a cost $c_{ij}$. Other inputs like capacity and time-window are concatenated with coordinates to formalize features of each node. Then, model starts to decode feasible solutions auto-regressively:

$$p_\theta(\tau|G) = \prod_{t=1}^{T} p_\theta(a_t|a_{t-1}, G), \quad (9)$$

where $a_t$, $\tau$ and $\theta$ represent next step's action, generated trajectory and model parameters, respectively. During decoding, remaining capacity, elapsed time and traveled distance are recorded and treated as dynamical features.

*Remarks.* Note that although some intermediate representations of our proposed module locate in non-Euclidean spaces, all of the problem instances from each considered task are grounded in Euclidean space, following the data generation process in (Zhou et al., 2024; Berto et al., 2024).

# 4. Methodology

In this section, we illustrate the mixed-curvature module using the architecture proposed in (Kwon et al., 2020; Liu et al., 2024) as an example. However, this module can be seamlessly integrated into other neural architectures, such as MVMoE(-L) (Zhou et al., 2024) and RouteFinder (Berto et al., 2024). As shown in Figure 2, our framework firstly embeds concatenated features of graphs by Euclidean embedders and then projects these features from flat space into a mixed-curvature space by partitioning the original feature space into multiple geometric subspaces, each with a learnable curvature parameter. Furthermore, to mitigate the geometry mismatch phenomena at each encoder layer, we interpolate representations between current layer and previous layer. By doing this, we can enable model to acquire a soft and learnable alignment process among incompatible geometric spaces, facilitating a smoother information flow and ultimately enhancing the quality of learned representations.

## 4.1. Mixed-Curvature Linear Transformation

In the first stage, we split original feature space with dimension $D$ into $C$ subspaces so that each subspace is equipped with a learnable curvature parameter $\kappa$ and a smaller dimension $\frac{D}{C}$. After conducting transformations in geometric spaces, these scattered features will be merged together to formalize a complete vector with original dimension $D$. Similar to (Ganea et al., 2018; Gu et al., 2018; Cho et al., 2023), we rely on *exponential map* and *logarithmic map* defined in Eq. (8) to perform feature transformation operations. To be specific, suppose that our intermediate feature representations $\mathbf{X}$ now reside in mixed-curvature space and we aim to perform operations such as feature transformation $f = \mathbf{X}\mathbf{W}$, then we have:

$$\hat{\mathbf{X}} = Exp_{\mathbf{0}}^{\overline{\kappa}_2}\left(Log_{\mathbf{0}}^{\overline{\kappa}_1}(\mathbf{X})\mathbf{W}\right), \quad (10)$$

where we choose the original point $\mathbf{0}$ to define the tangent space. Note that the tangent space is flat, so $\mathbf{W}$ actually resides in the Euclidean space, allowing us to train it with the standard optimizer like Adam. Also note that $\overline{\kappa}_1$ may not always equal to $\overline{\kappa}_2$. By applying these different curvatures, our proposed module can capture diverse geometric structures in a layer-by-layer manner instead of being confined to subspaces with limited semantics. Activation functions and normalization modules can be similarly adapted, following the form of Eq. (10).

## 4.2. Mixed-Curvature Augmented Embedding Layer

In the original POMO-MTL architecture (Liu et al., 2024), the embedding layer consists of two parts: one for the depot node and the other for the customer nodes. However, this setup can lead to suboptimal embeddings that fail to capture the full range of geometric information in the inputs. To ad-

dress this issue, we first project the depot and customer node embeddings from Euclidean space into a mixed-curvature space (i.e., the *embedding layer* in Figure 2), and then apply two independent mixed-curvature layers following Eq. (10). This process yields two feature representations, $\hat{\mathbf{X}}_{dep}$ and $\hat{\mathbf{X}}_{cus}$. We then concatenate them to form the final embedding:

$$\mathbf{X}_{emb} = [Log_{\mathbf{0}}^{\kappa}(\hat{\mathbf{X}}_{dep}); Log_{\mathbf{0}}^{\kappa}(\hat{\mathbf{X}}_{cus})], \quad (11)$$

where $[;]$ denotes the concatenation operation.

## 4.3. Mixed-Curvature Augmented Encoder Layer

In the prior experiments for our proposed architecture, we observe that naively propagating features from non-Euclidean subspaces (*e.g.*, hyperbolic or hyperspherical) into attention blocks often leads to performance degradation. We suspect this phenomenon is attributed to the shift of the receptive field with respect to the network depth: In shallow layers, the model primarily captures localized structural information (so the connectivity resembles a sparse graph which is a tree-like structure), where curvature tends to be negative (Nickel & Kiela, 2017; 2018). However, as the depth increases, the receptive field expands and the model begins to aggregate global information thus entangling the features of all nodes. This leads the model into hyperspherical spaces where points in feature space become substantially interconnected. Such properties of neural networks introduce a form of curvature mismatch between consecutive layers, leading into inferior performances. Drawing inspiration from the Mix-up technique (Zhang et al., 2018), which stabilizes training by interpolating representations of different samples, we design a similar mixing strategy that interpolates features across layers before sending them into current layer's attention block. Specifically, we formalize the input to each attention block as a weighted sum of the original Euclidean representation and the logarithmic-mapped mixed-curvature features from the previous layer:

$$\mathbf{X}^k = \alpha * \mathbf{X}^{k-1} + \beta * Log_{\mathbf{0}}^{\overline{\kappa}}(\hat{\mathbf{X}}^{k-1}), \quad (12)$$

where $\alpha, \beta$ are learnable parameters. In this way, model itself can gradually adjust to the evolving curvatures across layers. Unlike previous rigid transitions, this module encourages the model to retrieve previous layer's information in a dynamical manner, thereby improving the quality of representations.

## 4.4. Loss

Once decoder receives embeddings from previous established mixed-curvature encoder layers, the model starts to generate logits for each trajectory in the way of Eq. (9). Following (Kwon et al., 2020), we adopt the *reinforce* algorithm proposed in (Williams, 1992) for training. Specifically,

| Type | Model | n=50 Obj | Gap | Time | n=100 Obj | Gap | Time | Type | Model | n=50 Obj | Gap | Time | n=100 Obj | Gap | Time |
|---|---|---|---|---|---|---|---|---|---|---|---|---|---|---|---|
| CVRP | HGS | 10.334 | 0.000% | 4.6m | 15.504 | 0.000% | 9.1m | VRPTW | HGS | 14.509 | 0.000% | 8.4m | 24.339 | 0.000% | 19.6m |
| | LKH3 | 10.346 | 0.115% | 9.9m | 15.590 | 0.556% | 18.0m | | LKH3 | 14.607 | 0.664% | 5.5m | 24.721 | 1.584% | 7.8m |
| | OR-Tools | 10.540 | 1.962% | 10.4m | 16.381 | 5.652% | 20.8m | | OR-Tools | 14.915 | 2.694% | 10.4m | 25.894 | 6.297% | 20.8m |
| | OR-Tools(×10) | 10.418 | 0.788% | 1.7h | 15.935 | 2.751% | 3.5h | | OR-Tools(×10) | 14.665 | 1.011% | 1.7h | 25.212 | 3.482% | 3.5h |
| | POMO-MTL | 10.437 | 0.987% | 3s | 15.790 | 1.846% | 9s | | POMO-MTL | 15.032 | 3.637% | 3s | 25.610 | 5.313% | 12s |
| | Mixed-POMO-MTL | 10.436 | 0.980% | 5s | 15.771 | 1.731% | 14s | | Mixed-POMO-MTL | 15.021 | 3.556% | 4s | 25.556 | 5.090% | 12s |
| | MVMoE-L | 10.434 | 0.955% | 4s | 15.771 | 1.728% | 11s | | MVMoE-L | 15.013 | 3.500% | 4s | 25.519 | 4.927% | 14s |
| | Mixed-MVMoE-L | 10.431 | 0.933% | 6s | 15.758 | 1.645% | 14s | | Mixed-MVMoE-L | 15.002 | 3.421% | 4s | 25.506 | 4.872% | 15s |
| | MVMoE | 10.428 | 0.896% | 4s | 15.760 | 1.653% | 12s | | MVMoE | 14.999 | 3.410% | 4s | 25.512 | 4.903% | 15s |
| | Mixed-MVMoE | **10.424** | **0.865%** | 7s | **15.751** | **1.599%** | 16s | | Mixed-MVMoE | **14.995** | **3.373%** | 4s | **25.473** | **4.732%** | 16s |
| OVRP | LKH3 | 6.511 | 0.198% | 4.5m | 9.828 | 0.000% | 5.3m | VRPL | LKH3 | 10.571 | 0.790% | 7.8m | 15.771 | 0.000% | 16.0m |
| | OR-Tools | 6.531 | 0.495% | 10.4m | 10.010 | 1.806% | 20.8m | | OR-Tools | 10.677 | 1.746% | 10.4m | 16.496 | 4.587% | 20.8m |
| | OR-Tools(×10) | 6.498 | 0.000% | 1.7h | 9.842 | 0.122% | 3.5h | | OR-Tools(×10) | 10.495 | 0.000% | 1.5h | 16.004 | 1.444% | 3.5h |
| | POMO-MTL | 6.671 | 2.634% | 2s | 10.169 | 3.458% | 9s | | POMO-MTL | 10.513 | 0.201% | 2s | 15.846 | 0.479% | 10s |
| | Mixed-POMO-MTL | 6.670 | 2.637% | 3s | 10.154 | 3.312% | 10s | | Mixed-POMO-MTL | 10.511 | 0.185% | 3s | 15.827 | 0.362% | 11s |
| | MVMoE-L | 6.665 | 2.548% | 3s | 10.145 | 3.214% | 11s | | MVMoE-L | 10.506 | 0.131% | 3s | 15.821 | 0.323% | 12s |
| | Mixed-MVMoE-L | 6.658 | 2.448% | 4s | 10.136 | 3.133% | 12s | | Mixed-MVMoE-L | 10.502 | 0.098% | 3s | 15.813 | 0.270% | 13s |
| | MVMoE | 6.655 | 2.402% | 3s | 10.138 | 3.136% | 12s | | MVMoE | 10.501 | 0.092% | 3s | 15.812 | 0.261% | 14s |
| | Mixed-MVMoE | **6.651** | **2.336%** | 4s | **10.119** | **2.946%** | 12s | | Mixed-MVMoE | **10.497** | **0.052%** | 4s | **15.806** | **0.227%** | 14s |
| VRPB | OR-Tools | 8.127 | 0.989% | 10.4m | 12.185 | 2.594% | 20.8m | OVRPTW | OR-Tools | 8.737 | 0.592% | 10.4m | 14.635 | 1.756% | 20.8m |
| | OR-Tools(×10) | 8.046 | 0.000% | 1.7h | 11.878 | 0.000% | 3.5h | | OR-Tools(×10) | 8.638 | 0.000% | 1.7h | 14.380 | 0.000% | 3.5h |
| | POMO-MTL | 8.182 | 1.684% | 2s | 12.072 | 1.674% | 8s | | POMO-MTL | 8.987 | 3.470% | 3s | 15.008 | 4.411% | 12s |
| | Mixed-POMO-MTL | 8.179 | 1.645% | 2s | 12.043 | 1.427% | 8s | | Mixed-POMO-MTL | 8.982 | 3.420% | 3s | 14.948 | 3.996% | 12s |
| | MVMoE-L | 8.176 | 1.605% | 3s | 12.036 | 1.368% | 10s | | MVMoE-L | 8.974 | 3.322% | 4s | 14.940 | 3.941% | 14s |
| | Mixed-MVMoE-L | 8.170 | 1.531% | 3s | 12.025 | 1.265% | 10s | | Mixed-MVMoE-L | 8.964 | 3.219% | 4s | 14.911 | 3.749% | 15s |
| | MVMoE | 8.170 | 1.540% | 3s | 12.027 | 1.285% | 10s | | MVMoE | 8.964 | 3.210% | 4s | 14.927 | 3.852% | 15s |
| | Mixed-MVMoE | **8.164** | **1.456%** | 3s | **12.011** | **1.153%** | 11s | | Mixed-MVMoE | **8.950** | **3.060%** | 4s | **14.888** | **3.579%** | 16s |

*Table 1.* Performances on 6 seen tasks by following the setting of (Zhou et al., 2024). Each task is assigned with 1,000 unseen instances for testing. The best performances are annotated with bold and domains improved by our module are highlighted with underlines.

based on Eq. (9), our objective function is defined as:

$$\mathcal{L} = E_{\tau \sim p_\theta(\tau|G)}[R(\tau)], \quad (13)$$

and during the optimization stage, the gradient of the objective function takes the following form:

$$\nabla_\theta \mathcal{L} = \frac{1}{N} \sum_{i=1}^{N} \left( R(\tau^i) - b^i(G) \right) \nabla_\theta \log p_\theta(\tau^i|G), \quad (14)$$

where $R(\tau^i)$ denotes the reward (in our case, it is defined as the negative length) obtained from the $i$-th generated trajectory $\tau^i$, and $b^i(G)$ is the shared baseline introduced to reduce the variance in optimization stage. For other models such as MVMoE(-L) (Zhou et al., 2024), an additional objective may be added to balance the load among different expert modules.

*Remarks.* As noted in (Cho et al., 2023), the linear transformation defined in Eq. (10) is differentiable with respect to curvature $\kappa$. Hence, we can treat $\kappa$ as a learnable parameter and optimize it during training.

## 5. Experiments

In this section, we present our experimental findings to demonstrate the effectiveness of the proposed mixed-curvature pre-training paradigm in enabling a multi-task solver for vehicle routing problems (VRPs). Specifically, we evaluate our approach on 24 distinct VRP variants (or tasks) spanning 6 different constraint types. All experiments

are conducted on a machine equipped with four NVIDIA RTX A6000 GPUs, each with 48 GB of memory. In the following, we first introduce the baselines used in our experiments, then describe the training and testing configurations. Finally, we report the experimental results along with detailed result analysis[1].

### Baselines

The baselines used in our study fall into two categories: traditional heuristic solvers and neural solvers. Below, we provide specific details for each baseline:

**HGS (Vidal, 2022)**: A traditional solver based on genetic algorithm, designed to tackle different VRP variants.

**LKH3 (Helsgaun, 2017)**: A widely used heuristic algorithm for solving VRP variants. It employs a k-opt mechanism where, during the search stage, k edges are removed and reconnected to discover potentially better solutions.

**OR-Tools (Perron & Didier, 2024)**: A comprehensive solver developed by Google that supports various combinatorial optimization tasks, including VRPs.

**POMO-MTL (Liu et al., 2024)**: A multi-task extension of POMO (Kwon et al., 2020), which enables the model to address multiple VRPs simultaneously.

**MVMoE(-L) (Zhou et al., 2024)**: MVMoE incorporates

---

[1]Our code is available at: `https://github.com/lsyysl9711/Mixed_Curvature_VRPs`

| Type | Model | Obj | n=50 Gap | Time | Obj | n=100 Gap | Time | Type | Model | Obj | n=50 Gap | Time | Obj | n=100 Gap | Time |
|---|---|---|---|---|---|---|---|---|---|---|---|---|---|---|---|
| OVRPB | OR-Tools | 5.764 | 0.332% | 10.4m | 8.522 | 1.852% | 20.8m | OVRPL | OR-Tools | 6.522 | 0.480% | 10.4m | 9.966 | 1.783% | 20.8m |
| | OR-Tools(×10) | 5.745 | 0.000% | 1.7h | 8.365 | 0.000% | 3.5h | | OR-Tools(×10) | 6.490 | 0.000% | 1.7h | 9.790 | 0.000% | 3.5h |
| | POMO-MTL | 6.116 | 6.430% | 2s | 8.979 | 7.335% | 8s | | POMO-MTL | 6.668 | 2.734% | 2s | 10.126 | 3.441% | 10s |
| | Mixed-POMO-MTL | 6.112 | 6.348% | 3s | 9.021 | 7.831% | 9s | | Mixed-POMO-MTL | 6.667 | 2.708% | 3s | 10.116 | 3.350% | 11s |
| | MVMoE-L | 6.122 | 6.522% | 3s | 8.972 | 7.243% | 10s | | MVMoE-L | 6.659 | 2.597% | 3s | 10.106 | 3.244% | 12s |
| | Mixed-MVMoE-L | 6.102 | 6.175% | 3s | 8.951 | 6.997% | 11s | | Mixed-MVMoE-L | 6.653 | 2.497% | 4s | 10.098 | 3.159% | 13s |
| | MVMoE | 6.092 | 5.999% | 3s | 8.959 | 7.088% | 11s | | MVMoE | 6.650 | 2.454% | 3s | 10.097 | 3.148% | 13s |
| | Mixed-MVMoE | **6.084** | **5.871%** | 4s | **8.934** | **6.800%** | 12s | | Mixed-MVMoE | **6.648** | **2.419%** | 4s | **10.079** | **2.971%** | 14s |
| VRPBL | OR-Tools | 8.131 | 1.254% | 10.4m | 12.905 | 2.586% | 20.8m | VRPBTW | OR-Tools | 15.053 | 1.857% | 10.4m | 26.217 | 2.858% | 20.8m |
| | OR-Tools(×10) | 8.029 | 0.000% | 1.7h | 11.790 | 0.000% | 3.5h | | OR-Tools(×10) | 14.771 | 0.000% | 1.7h | 25.496 | 0.000% | 3.5h |
| | POMO-MTL | 8.188 | 1.971% | 2s | 11.998 | 1.793% | 9s | | POMO-MTL | 16.055 | 8.841% | 3s | 27.319 | 7.413% | 11s |
| | Mixed-POMO-MTL | 8.182 | 1.905% | 3s | 11.964 | 1.514% | 10s | | Mixed-POMO-MTL | 16.071 | 8.943% | 3s | 27.327 | 7.457% | 12s |
| | MVMoE-L | 8.180 | 1.872% | 3s | 11.960 | 1.473% | 10s | | MVMoE-L | 16.041 | 8.745% | 3s | 27.265 | 7.190% | 13s |
| | Mixed-MVMoE-L | 8.172 | 1.781% | 3s | 11.949 | 1.378% | 12s | | Mixed-MVMoE-L | 16.039 | 8.715% | 4s | 27.223 | 7.018% | 11s |
| | MVMoE | 8.172 | 1.776% | 3s | 11.945 | 1.346% | 11s | | MVMoE | 16.022 | 8.600% | 3s | 27.236 | 7.078% | 14s |
| | Mixed-MVMoE | **8.168** | **1.729%** | 4s | **11.936** | **1.264%** | 12s | | Mixed-MVMoE | **16.014** | **8.545%** | 4s | **27.208** | **6.967%** | 15s |
| VRPLTW | OR-Tools | 14.815 | 1.432% | 10.4m | 25.823 | 2.534% | 20.8m | OVRPBL | OR-Tools | 5.771 | 0.549% | 10.4m | 8.555 | 2.459% | 20.8m |
| | OR-Tools(×10) | 14.598 | 0.000% | 1.7h | 25.195 | 0.000% | 3.5h | | OR-Tools(×10) | 5.739 | 0.000% | 1.7h | 8.348 | 0.000% | 3.5h |
| | POMO-MTL | 14.961 | 2.586% | 3s | 25.619 | 1.920% | 13s | | POMO-MTL | 6.104 | 6.306% | 2s | 8.961 | 7.343% | 9s |
| | Mixed-POMO-MTL | 14.966 | 2.621% | 3s | 25.561 | 1.673% | 14s | | Mixed-POMO-MTL | 6.102 | 6.282% | 3s | 9.009 | 7.919% | 10s |
| | MVMoE-L | 14.953 | 2.535% | 4s | 25.529 | 1.545% | 16s | | MVMoE-L | 6.104 | 6.310% | 3s | 8.957 | 7.300% | 11s |
| | Mixed-MVMoE-L | 14.941 | 2.448% | 4s | 25.521 | 1.515% | 17s | | Mixed-MVMoE-L | 6.090 | 6.077% | 3s | 8.935 | 7.027% | 11s |
| | MVMoE | 14.937 | 2.421% | 4s | 25.514 | 1.471% | 17s | | MVMoE | 6.076 | 5.843% | 3s | 8.942 | 7.115% | 12s |
| | Mixed-MVMoE | **14.931** | **2.387%** | 4s | **25.486** | **1.365%** | 18s | | Mixed-MVMoE | **6.068** | **5.705%** | 4s | **8.920** | **6.857%** | 12s |
| OVRPBTW | OR-Tools | 8.758 | 0.927% | 10.4m | 14.713 | 2.268% | 20.8m | OVRPLTW | OR-Tools | 8.728 | 0.656% | 10.4m | 14.535 | 1.779% | 20.8m |
| | OR-Tools(×10) | 8.675 | 0.000% | 1.7h | 14.384 | 0.000% | 3.5h | | OR-Tools(×10) | 8.669 | 0.000% | 1.7h | 14.279 | 0.000% | 3.5h |
| | POMO-MTL | 9.514 | 9.628% | 3s | 15.879 | 10.453% | 10s | | POMO-MTL | 8.987 | 3.633% | 3s | 14.896 | 4.374% | 12s |
| | Mixed-POMO-MTL | 9.523 | 9.734% | 3s | 15.844 | 10.192% | 11s | | Mixed-POMO-MTL | 8.984 | 3.600% | 3s | 14.845 | 4.020% | 12s |
| | MVMoE-L | 9.515 | 9.630% | 3s | 15.841 | 10.188% | 12s | | MVMoE-L | 8.974 | 3.488% | 4s | 14.839 | 3.971% | 14s |
| | Mixed-MVMoE-L | 9.506 | 9.530% | 4s | 15.802 | 9.899% | 15s | | Mixed-MVMoE-L | 8.961 | 3.335% | 4s | 14.816 | 3.816% | 15s |
| | MVMoE | 9.486 | 9.308% | 4s | 15.808 | 9.948% | 13s | | MVMoE | 8.966 | 3.396% | 4s | 14.828 | 3.903% | 15s |
| | Mixed-MVMoE | **9.483** | **9.283%** | 4s | **15.779** | **9.749%** | 14s | | Mixed-MVMoE | **8.951** | **3.225%** | 4s | **14.779** | **3.560%** | 16s |
| VRPBLTW | OR-Tools | 14.890 | 1.402% | 10.4m | 25.979 | 2.518% | 20.8m | OVRPBLTW | OR-Tools | 8.729 | 0.624% | 10.4m | 14.496 | 1.724% | 20.8m |
| | OR-Tools(×10) | 14.667 | 0.000% | 1.7h | 25.342 | 0.000% | 3.5h | | OR-Tools(×10) | 8.673 | 0.000% | 1.7h | 14.250 | 0.000% | 3.5h |
| | POMO-MTL | 15.980 | 9.035% | 3s | 27.247 | 7.746% | 12s | | POMO-MTL | 9.532 | 9.851% | 3s | 15.738 | 10.498% | 11s |
| | Mixed-POMO-MTL | 15.998 | 9.139% | 3s | 27.219 | 7.658% | 13s | | Mixed-POMO-MTL | 9.541 | 9.946% | 3s | 15.720 | 10.358% | 13s |
| | MVMoE-L | 15.963 | 8.915% | 4s | 27.177 | 7.473% | 14s | | MVMoE-L | 9.518 | 9.682% | 4s | 15.706 | 10.263% | 13s |
| | Mixed-MVMoE-L | 15.961 | 8.871% | 4s | 27.129 | 7.278% | 15s | | Mixed-MVMoE-L | 9.509 | 9.582% | 4s | 15.673 | 10.027% | 14s |
| | MVMoE | 15.945 | 8.775% | 4s | 27.142 | 7.332% | 15s | | MVMoE | 9.503 | 9.516% | 4s | 15.671 | 10.009% | 14s |
| | Mixed-MVMoE | **15.932** | **8.690%** | 4s | **27.136** | **7.304%** | 16s | | Mixed-MVMoE | **9.498** | **9.462%** | 4s | **15.636** | **9.772%** | 16s |

*Table 2.* Performances on 10 unseen tasks following the setting of (Zhou et al., 2024). Each task is assigned with 1,000 instances for testing. The best performances are annotated with bold and domains improved by our module are highlighted with underlines.

mixture-of-expert (MoE) modules into both encoder and decoder layers, differing from the original POMO-MTL architecture. In the meanwhile, MVMoE-L is a lightweight variant of MVMoE that accelerates the routing mechanism while maintaining computational efficiency.

**RF-X (Berto et al., 2024)**: RouteFinder (or its variant) offers a more fine-grained feature fusion approach that further enhances performances of POMO-MTL, and MVMoE(-L).

### Training Configurations

Due to the significant differences in experimental settings between (Zhou et al., 2024) and (Berto et al., 2024), we divide our experiments into two parts. The first part strictly follows the training configurations outlined in (Zhou et al., 2024), and the analysis of these results is presented in Section 5.1. The second part follows the experimental setup from (Berto et al., 2024), with the corresponding analysis provided in Section 5.2.

*Configurations with (Zhou et al., 2024).* We have two prob-

lem scales: 50 and 100 nodes in each instance. As mentioned earlier, our pre-training paradigm can be seamlessly integrated into any existing architectures, so we take POMO-MTL (Liu et al., 2024), and MVMoE(-L) (Zhou et al., 2024) as our backbones. We adopt Adam as our optimizer. The learning rate, weight decay and batch size are set to 1e-4 and 1e-6 and 128, respectively. We train each model with 5,000 epochs and for each epoch there are 20,000 instances. During the last 500 epochs, we decay the learning rate by 10. At the very beginning, we initialize all of the curvatures as 0 and jointly optimize them with other parameters. Note that only 6 VRP variants are used for training. Further details about hyper-parameters are listed in Table 12, Appendix.2.

*Configurations with (Berto et al., 2024).* The problem scales consist of 50 and 100 as well. We take RF-X (Berto et al., 2024) as the backbone, and follow the settings in its original paper, where each model is only trained with 300 epochs and each epoch is assigned with 100,000 training instances. Note that different from (Zhou et al., 2024), in this case, 16 VRP tasks are all used for training RF-X. Besides, the

learning rate, weight decay and batch size are set to 3e-4, 1e-6 and 256, respectively. In epoch 270 and 295, we decay the learning rate by 10. The detailed experimental configurations of RF-X can be found in Table 17, Appendix.4.

**Validation Configurations**

We conduct four types of validation experiments: the in-distribution testing, zero-shot testing, few-shot testing, and real-world testing. We divide these evaluations into two sets of configurations corresponding to (Zhou et al., 2024) and (Berto et al., 2024).

*Configurations with (Zhou et al., 2024).* For each VRP task, we pre-collect 1,000 unseen instances and report gaps relative to the optimal (or best) known solutions. Following (Kwon et al., 2020; Zhou et al., 2024), we apply greedy rollout with 8× instance augmentation for fair comparisons, where best solutions for each instance are obtained by solving multiple (8×) equivalent instances. Those equivalent instances are acquired by rotating or clipping the original instances (Kwon et al., 2020). The in-distribution test comprises 6 VRP tasks included during training, while the zero-shot test includes 10 tasks not seen during training. For few-shot testing, we choose VRPBLTW and OVRPBLTW to assess model performance in low-data scenarios. Lastly, we follow (Zhou et al., 2024) for real-world evaluations on set-X (Uchoa et al., 2017) for CVRP and set-Solomon (Solomon, 1987) for VRPTW.

*Configurations with (Berto et al., 2024).* In this setting, each task is again assigned 1,000 unseen instances, with the gaps to the optimal (or best) solutions reported. 16 VRP tasks are designated as the seen ones in the in-distribution test, while 8 tasks are the unseen ones in few-shot evaluations. For real-world testing, we follow (Berto et al., 2024) and use sets A, B, E, F, M, P, and X from CVRPLib (Uchoa et al., 2017) to assess the model performance under more practical conditions.

*Distortion Rate and Curvature Analysis.* Apart from the above validations, we also analyze the distortion rates and visualize the learned curvatures. Due to space limits, we move them to Appendix.1.

## 5.1. Results Compared with (Zhou et al., 2024)

**In-distribution Test on Seen Tasks**. To evaluate performance on tasks seen during training, we begin by testing the models on 6 such tasks. The results, presented in Table 1, indicate that the original MVMoE augmented with the mixed-curvature module outperforms all prior baselines. Furthermore, the mixed-curvature module improves POMO-MTL and MVMoE-L performance on 5 out of 6 tasks and 6 out of 6 tasks, respectively, emphasizing the general benefits of incorporating geometric subspaces at the pre-training

stage. Notably, in the more challenging scenario where $N = 100$, MVMoE-L with the mixed-curvature module outperforms the original MVMoE across all 6 tasks, demonstrating strong versatility across different problem sizes.

**Zero-shot Test on Unseen Tasks**. To further assess the zero-shot predictive capabilities of our approach, we evaluate each model on 10 tasks that were not included in training. The results, shown in Table 2, reveal that MVMoE with the mixed-curvature module achieves state-of-the-art performance on all 10 tasks at both node number scales. This outcome underscores the effectiveness of our module in enabling solvers to generalize to previously unseen scenarios.

**Few-Shot Test**. Following the experimental setup of (Zhou et al., 2024), we examine two previously unseen tasks, OVRPBLTW and VRPBLTW, to gauge each model's performance in a few-shot context. Specifically, we fine-tune each model for 10 epochs, with each epoch drawing on 10,000 randomly sampled training instances. As illustrated in Figure 3, Mixed-MVMoE and Mixed-MVMoE-L outperform the baseline models, demonstrating that the incorporation of mixed-curvature spaces can enhance performance in low-resource settings as well.

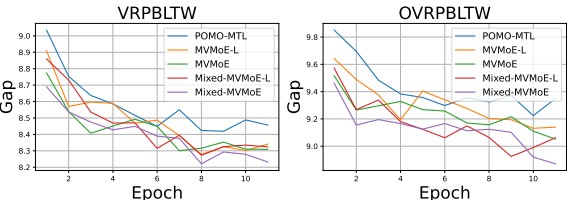

*Figure 3.* The few-shot performance on two unseen tasks following settings of (Zhou et al., 2024). The x and y axis represent epochs and gaps, respectively. Here each problem instance has 50 nodes.

**Real-World Instances Test**. We also evaluate the models on real-world testing instances sourced from CVRPLib, categorized broadly into moderate-scale and large-scale settings. Table 13, Table 14 and Table 15, Appendix.3 show that models incorporating our mixed-curvature module outperform both the single-task model (POMO) and multi-task models (POMO-MTL and MVMoE). The results indicate not only a reduction in the performance gap on moderate-scale problems but also a consistent narrowing of the gap on large-scale instances, showing our mixed-curvature module enables the original model to adapt effectively to real-world scenarios.

## 5.2. Results Compared with (Berto et al., 2024)

In RouteFinder (Berto et al., 2024), all of the 16 tasks from (Zhou et al., 2024) are used for training, and the results are shown in Table 18. From these presented outcomes, it is evident that RF-TE combined with the mixed-curvature

module achieves the lowest performance gaps on 16 of the 16 tasks on both node sizes. Besides, augmented with mixed-curvature modules, backbones like MTPOMO, MV-MoE and RF-MVMoE get consistent improvements on their performances. We also evaluate its performance on 7 real-world benchmarks from CVRPLib, where RF-TE equipped with the mixed-curvature module further reduces the average gaps (as shown in Table 19). Moreover, we assess its capabilities on 8 few-shot tasks. As illustrated in Table 20, the mixed-curvature-based model trained with EAL (shorted for *Efficient Adapter Layer* in (Berto et al., 2024)) significantly surpasses the performance of the original model tuned with EAL. Furthermore, when trained from scratch, the mixed-curvature-based model outperforms the original model with EAL on 6 of the 8 tasks, demonstrating the effectiveness of the mixed-curvature module in enhancing existing multi-task VRP solvers.

### 5.3. More Ablation Studies and Discussions

**Effects of Increased Parameters.** Since we insert several mixed-curvature modules into the embedder and encoder layer, the total number of parameters is increased and we list the number of parameters of each model in Table 9. Compared to previous baselines, the increased ratio is between 3.57% and 10.56%. Moreover, from results presented in Table 7, we can observe that only increasing the number of parameters will downgrade performances in most cases. In specifics, we replace the mixed-curvature space modules in encoder with their Euclidean counterparts so that the number of parameters is still in the same level as before (we name these models as Euc-POMO-MTL, Euc-MVMoE-L and Euc-MVMoE). For the Euc-POMO-MTL, it achieves the worst performances on 13 out of 16 and 11 out of 16 tasks with node size of 50 and 100, respectively. For the Euc-MVMoE, it achieves worst performances on 12 out of 16 tasks under both of the node size settings. These evidences demonstrate that naively increasing the number of parameters will often lead into inferior results in most cases, which indicates that the improvements on performances largely benefit from the introduction of mixed-curvature spaces.

**Effects of Mix-up Modules.** In this part, we discuss the effectiveness of Mix-up modules and how will it affect the performances of model. We utilize Mixed-POMO-MTL, Mixed-MVMoE(-L) to conduct ablation experiments. From results presented in Table 8, we can observe that after removing the Mix-up modules from the encoder, the performances for Mixed-POMO-MTL will become worse on 12 out of 16 tasks with node size $N = 50$ and 11 out of 16 tasks with node size $N = 100$, respectively. Similarly, the Mixed-MVMoE without Mix-Up modules will lose their SOTA performances on 11-12 tasks across problem types and node sizes. These demonstrate that the Mix-up modules can further enhance performances, which validates this mod-

ule's utility and necessity in smoothing transitions between different curvature spaces.

**Effects of the Number of Subspaces.** In our experiment, we set the number of subspaces in mixed-curvature space as 8 and each subspace is assigned with 16 feature dimensions. To further investigate the effects of the number of subspaces on model's performances, we also try 4 and 16 in Mixed-POMO-MTL to illustrate the difference. We show results on both node scales in Table 10. From the presented results for $N = 50$, we can find that the Mixed-POMO-MTL-8 can achieve the best averaged performances on 16 tasks, while the Mixed-POMO-MTL-4 and Mixed-POMO-MTL-16 achieve relatively inferior performances. We guess the reason is that although POMO-MTL-Mixed-4 acquires larger subspaces, its diversity is severely limited by the number of subspaces compared to the other two. In the meanwhile, Mixed-POMO-MTL-16 enhances its diversity but the feature dimension maybe too small to capture important geometric information from the inputs. However, when problem size becomes larger, Mixed-POMO-MTL-16 can surpass Mixed-POMO-MTL-4 by a large margin and greatly shorten the gap with respect to Mixed-POMO-MTL-4.

**Discussions of Running Time.** Since several mixed-curvature modules are inserted into the embedder and encoder, the running time will be increased. As reflected in Table 1 and Table 2, the model requires more time to process instances on almost every VRPs task. Taking $N = 100$ as an example, Mixed-POMO-MTL, Mixed-MVMoE-L and Mixed-MVMoE introduce 10.72%, 7.68%, 7.56% extra time costs to their own backbones, respectively. These show that although mixed-curvature modules bring extra computational burdens, the added costs are moderate.

## Conclusions

In this work, we present a novel pre-training paradigm that processes features in curved geometric spaces for solving multi-task VRPs. By splitting the original feature space into multiple subspaces, each with its own learnable curvature, we enable the model to capture diverse geometric structures from inputs. Extensive experiments show that our mixed-curvature modules consistently enhance various backbone architectures, highlighting the promise of mixed-curvature spaces in improving multi-task VRP solvers. However, our approach has limitations. First, the frequent use of exponential and logarithmic map operations introduces extra time costs and may cause some unstable numerical phenomena. Second, our study does not consider large-scale instances with around 10,000 nodes which has achieved great attentions recently. Thirdly, we allocate 16 dimensions for each curvature subspace but other adaptive methods like neural-architecture-search (Elsken et al., 2019) may also be feasible. We plan to explore and address them in future works.

## Impact Statement

This paper presents work whose goal is to advance the field of deep learning for vehicle routing problems. To our best knowledge, there is no potential societal consequence of our work since the module developed in our work focuses on facilitating the constrained optimizations.

## Acknowledgment

This research/project is supported by the National Research Foundation, Singapore under its AI Singapore Programme (AISG Award No: AISG3-RP-2022-031), the MTI under its AI Centre of Excellence for Manufacturing (AIMfg) (Award W25MCMF014), and the College of Computing and Data Science, Nanyang Technological University. We also want to express our sincere thanks to all reviewers and area chair for their constructive engagement and valuable suggestions during the rebuttal stage.

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

# A. Appendix

## Appendix.1. Definitions, Formulas and Visualizations

### Definition of Ollivier-Ricci Curvatures

Curvature describes how much a curve deviates from being a straight line or how much a surface bends in the space. The Ollivier-Ricci curvature (Ollivier, 2009) is a discrete notion of curvature that extends Ricci curvature from smooth (thus continuous) Riemannian manifolds to structures like graphs and networks (Fu et al., 2025). To be specific, suppose that we are given a metric space denoted by $(X, d)$ where $d$ represents metric distance (in our case $X$ consists of two dimensions (coordinates) and $d$ is the L2 distance between two points), then the Ollivier-Ricci curvature has the following form:

$$\kappa(x, y) = 1 - \frac{W(\mu_x, \mu_y)}{d(x, y)}, \quad x, y \in X \tag{15}$$

where $W(\cdot, \cdot)$ denotes Wasserstein distance between two probability measures:

$$\mu_x(z) = \frac{c_{xz}}{\sum_{i=1}^{N_x} c_{xi}}, \quad \mu_y(z) = \frac{c_{yz}}{\sum_{i=1}^{N_y} c_{yi}} \tag{16}$$

where $c_{xz}, c_{yz}$ are edge weights on edges $xz, yz$ and $N_x, N_y$ are numbers of adjacent neighbors of $x, y$, respectively. Based on these quantities, $\mu_x$ and $\mu_y$ actually measure the transition probability of the random walk starts from $x, y$ and ends at some point $z$. In other words, the smaller (bigger) the $W(\mu_x, \mu_y)$, the higher (lower) the chances that $x$ and $y$ will encounter each other within a few steps, which shares great similarity with the contraction (divergence) behaviors in spherical (hyperbolic) spaces. In our implementation, for each node $x \in X$, we use K-Nearest Neighbors algorithm to sample 5 nodes from each node's neighbour set and calculate the node curvature by averaging Ollivier-Ricci curvatures on the selected 5 edges. Finally, we can get the graph curvature by averaging all of these nodes' curvatures.

*Remarks.* Note that although we only take 5 edges for each node in the stage of curvature calculations, the embedder takes the complete graph as the input. The reason for us to choose KNN graphs to analyze curvatures is that during the delivery, not all of the edges are worth of being considered: some maybe less important and some maybe blocked due to hard constraints (*e.g.*, the same edge can't be visited twice during the delivery). As a result of this, applying KNN graphs here is more faithful to actual scenarios. Similar operations also appear in some prior works for solving traveling salesman problem (Joshi et al., 2019).

In previous works such as (Gu et al., 2018; Bachmann et al., 2020), the Parallelogram Law has been utilized to analyze the deviations of datasets from Euclidean geometry. This classical geometry identity checks whether quadruple of points behaves flat in the normal Euclidean sense: If the sum of the squares of the diagonals is greater (less) than the sum of the squares of the sides in a parallelogram then it indicates that data points are located in negative (positive) curvature spaces. The reasons for us to choose Ollivier-Ricci curvature mainly lie in two sides: 1) Enumerating all of the four point configurations from a graph with $N = 50$ nodes is computationally expensive while Ollivier-Ricci curvature offers a more light-weight method. 2) Ollivier-Ricci curvature enables a more nuanced understanding of geometric structures in datasets under optimal transport: it captures the degree to which local neighborhoods contract or diverge and this makes it especially well-suited for datasets represented as graphs or networks. For a more comprehensive comparison between these different curvature concepts, please refer to Table 4.

### Hyperbolic Model ($\kappa < 0$)

Let $\mathbf{x}, \mathbf{y}$ be a point on the manifold $\mathbb{H}(\kappa)$, $\mathbf{v} \in T_\mathbf{x}\mathbb{H}(\kappa)$ a tangent space vector and $\lambda_\mathbf{x} = \frac{2}{1 + \kappa \|\mathbf{x}\|_2^2}$. $\oplus_\kappa$ is defined in Eq. (3).

**Exponential Map:**

$$Exp_\mathbf{x}^\kappa(\mathbf{v}) = \mathbf{x} \oplus_\kappa \left( \tanh \left( \sqrt{-\kappa} \frac{\lambda_\mathbf{x}|\mathbf{v}|}{2} \right) \frac{\mathbf{v}}{|\mathbf{v}|} \right) \tag{17}$$

**Logarithmic Map:**

$$Log_\mathbf{x}^\kappa(\mathbf{y}) = \frac{2}{\lambda_\mathbf{x} \sqrt{-\kappa}} \tanh^{-1} \left( \sqrt{-\kappa} |\mathbf{x} \oplus_\kappa (-\mathbf{y})| \right) \cdot \frac{\mathbf{x} \oplus_\kappa (-\mathbf{y})}{|\mathbf{x} \oplus_\kappa (-\mathbf{y})|} \tag{18}$$

**Hyperspherical Model ($\kappa > 0$)**
Let $\mathbf{x}, \mathbf{y}$ be a point on the manifold $\mathbb{S}(\kappa)$, and $\mathbf{v} \in T_{\mathbf{x}}\mathbb{S}(\kappa)$ a tangent space vector. $d_{\kappa}(\mathbf{x}, \mathbf{y})$ is defined in Eq. (6).

**Exponential Map:**

$$Exp_{\mathbf{x}}^{\kappa}(\mathbf{v}) = \cos(\sqrt{\kappa}|\mathbf{v}|) \cdot \mathbf{x} + \sin(\sqrt{\kappa}|\mathbf{v}|) \cdot \frac{\mathbf{v}}{|\mathbf{v}|} \tag{19}$$

**Logarithmic Map:**

$$Log_{\mathbf{x}}^{\kappa}(\mathbf{y}) = d_{\kappa}(\mathbf{x}, \mathbf{y}) \cdot \frac{\mathbf{y} - \cos(d_{\kappa}(\mathbf{x}, \mathbf{y})) \cdot \mathbf{x}}{|\mathbf{y} - \cos(d_{\kappa}(\mathbf{x}, \mathbf{y})) \cdot \mathbf{x}|} \tag{20}$$

**Curvature Visualizations for another 10 VRPs Tasks**
The visualizations of curvatures for remaining datasets are included in Figure 4. Note that these 10 tasks are unseen during training stage, following the settings of (Zhou et al., 2024) while in (Berto et al., 2024), all of the previously mentioned 16 VRP variants are used for training the neural solvers. From Figure 4, we can observe that non-Euclidean information exists widely across these tasks, which further validates the necessity of introducing mixed-curvature space into neural solvers.

**Definition of Distortion Rate and Average Distortion Rate**
We adopt the notations from (Gu et al., 2018) where $U_1$ and $U_2$ denote two metric spaces (possibly with different dimensions) and they are equipped with distances $d_{U_1}(\cdot, \cdot)$ and $d_{U_2}(\cdot, \cdot)$, respectively. A mapping denoted by $f : U_1 \rightarrow U_2$ typically exists between these two spaces, such as deep neural networks. For any pair of points $a$ and $b$, the distortion rate induced from mapping $f$ is defined as:

$$\left| \frac{d_{U_1}(f(a), f(b))}{d_{U_2}(a, b)} - 1 \right|. \tag{21}$$

To evaluate the distortion rate globally, we consider all pairs of points and compute the average distortion as:

$$D_{avg} = \frac{1}{N} \sum_{a \neq b} \left| \frac{d_{U_1}(f(a), f(b))}{d_{U_2}(a, b)} - 1 \right|, \tag{22}$$

where $N$ denotes the number of node pairs such that $a \neq b$ without repetitions. By using Eq. (21) and (22), we can know how far away distances in the feature space learned by neural solvers deviates from those distances in the original input graphs. The lower the average distortion rate, the better the quality of representations we get from those models.

**Distortion Rate Analysis for Features of Encoder Module**
As previously mentioned, learning solely in Euclidean space can significantly distort distance information, adversely affecting the model's decision-making process and the final performances. In Table 3, we compare the distortion rates (defined in Eq. (22)) across various models. From the presented results, our approach achieves considerably lower distortion rates than the baselines, indicating that mixed-curvature space preserves original distances more faithfully. This allows the model to retain more accurate distance-related information, thus enhancing its decision-making capability.

| Model | Distortion |
|---|---|
| POMO-MTL | 2477.725 |
| Mixed-POMO-MTL | 1678.605 |
| MVMoE-L | 2923.151 |
| Mixed-MVMoE-L | 1981.076 |
| MVMoE | 2083.015 |
| Mixed-MVMoE | 1274.142 |

*Table 3.* Distortion rates of different models. We extract intermediate representations the final layer of encoder module and we use 1,000 CVRP instances with size $N = 50$ from testing datasets to calculate the distortion rate (defined in Eq. (22)).

**Curvature Analysis for Subspaces of Each Layer in Encoder module**

For the model familys in (Zhou et al., 2024), we visualize the curvature of each subspace in every encoder layer of Mixed-POMO-MTL (Figure 6, Appendix.3), MVMoE-L (Figures 7, Appendix.3), and MVMoE (Figures 8,Appendix.3). The distinct color gradients reveal that subspaces in shallower layers tend to lie in hyperbolic geometry. This observation aligns with the intuition that shallow layers primarily capture local structures, which often resemble trees or sparse graphs structures that naturally associated with negative curvatures. In contrast, as we move to deeper layers, the subspaces gradually transition towards the spherical geometry, reflecting the tendency of deeper layers to encode global information, where node features become increasingly aggregated or even collapsed with each other. For RF-X (Berto et al., 2024), we also visualize the curvature of subspaces in the model RF-MVMoE (Figure 9, Appendix.4) and RF-TE (Figure 10, Appendix.4), where we can observe similar curvature evolving patterns across layers.

| Curvature Type | Definition | Formula |
|---|---|---|
| Riemann Curvature Tensor (Lee, 2006) | Measures the failure of second covariant derivatives to commute with each other, encoding the intrinsic curvature of a Riemannian manifold. | $R(u,v)w = \nabla_u \nabla_v w - \nabla_v \nabla_u w - \nabla_{[u,v]} w$ |
| Ricci Curvature (Chow & Knopf, 2004) | Trace of the Riemann curvature tensor, denoting the average sectional curvature along different directions. | $\text{Ric}(u) = \sum_i R(u, e_i) e_i$ |
| Gaussian Curvature (Postnikov, 2013) | Product of the principal curvatures at a point on a surface, encoding the intrinsic measure of curvature. | $K = k_1 \cdot k_2$ |
| Mean Curvature (Postnikov, 2013) | Average of the principal curvatures. | $H = \frac{1}{2}(k_1 + k_2)$ |
| Principal Curvatures (Postnikov, 2013) | Maximum and minimum normal curvatures at a point on a surface. | Eigenvalues $k_1$, $k_2$ of Weingarten map (Cao et al., 2021) |
| Sectional Curvature (Lee, 2006) | It measures how the manifold curves in the direction of a tangent space. It generalizes Gaussian curvature into higher dimensions. | $K(u,v) = \frac{\langle R(u,v)v, u \rangle}{\|u \wedge v\|^2}$ |
| Ollivier-Ricci Curvature (Ollivier, 2009) | Measures the difference between two metric measure spaces based on optimal transport, especially suitable for discrete structures like graphs and networks. | $\kappa(x,y) = 1 - \frac{W(\mu_x, \mu_y)}{d(x,y)}$ |
| Parallelogram Law (Gu et al., 2018) | Whether the sum of the squares of the diagonals equals the sum of the squares of the sides in a parallelogram. | $\|x+y\|^2 + \|x-y\|^2 = 2\|x\|^2 + 2\|y\|^2$ |

*Table 4.* Different types of curvatures and their expressions in differential/Riemannian geometry. One major difference between Ollivier-Ricci curvature and Ricci/Gaussian/Mean/Principle curvatures is that Ollivier-Ricci curvature can handle discrete structures like graphs and networks while the others require the underlining manifold is continuous and smooth. Apart from Ollivier-Ricci curvature, there have been some other recent efforts that adapt curvatures on continuous spaces into discrete structures. For more information about this, we refer interested readers to (Najman & Romon, 2017).

| Acronym | Meaning |
| --- | --- |
| POMO | Policy Optimization with Multiple Optima for Reinforcement Learning (Kwon et al., 2020) |
| POMO-MTL | Policy Optimization with Multiple Optima for Reinforcement Learning with Multi-Task-Learning (Liu et al., 2024) |
| MVMoE-L | Multi-Task Vehicle Routing Solver with Mixture-of-Experts-Light (Zhou et al., 2024) |
| MVMoE | Multi-Task Vehicle Routing Solver with Mixture-of-Experts (Zhou et al., 2024) |
| RF | RouteFinder (Berto et al., 2024) |
| LKH | Lin-Kernighan-Helsgaun (Lin & Kernighan, 1973; Helsgaun, 2017) |
| HGS | Hybrid Genetic Search (Vidal, 2022) |
| HGS-PyVRP | A Python implementation of HGS for VRPs (Wouda et al., 2024) |
| VRP(s) | Vehicle Routing Problem(s) |
| C | Capacity |
| O | Open Route |
| L | Duration Limits |
| B | Backhauls |
| TW | Time Window |
| Mixed-X | The model named X augmented with Mixed-Curvature space modules |
| Euc-X | The model named X augmented with Euclidean space modules |

*Table 5.* List of acronyms that appear in the paper.

| Notations | Meaning |
| --- | --- |
| $\mathbf{x}, \mathbf{y}, \mathbf{v}$ | Finite dimensional vectors |
| $\mathcal{M}$ | Riemannian manifold |
| $g_{\mathbf{x}}$ | Riemannian metric |
| $\lambda_{\mathbf{x}}$ | Conformal factor of hyperbolic space |
| $\kappa$ | Curvature |
| $\overline{\kappa}$ | Consists of $(\kappa_1, \kappa_2, \ldots, \kappa_n)$, each entry corresponds to the curvature of a geometric space |
| $\overline{\mathbf{x}}$ | Consists of $(\mathbf{x}_1, \mathbf{x}_2, \ldots, \mathbf{x}_n)$ each entry corresponds to a chunk of feature located in a geometric space |
| $\mathbb{H}(\kappa)$ | Hyperbolic space with curvature $\kappa$ |
| $\mathbb{S}(\kappa)$ | Hyperspherical space with curvature $\kappa$ |
| $T_{\mathbf{x}}\mathcal{M}, T_{\mathbf{x}}\mathbb{H}(\kappa), T_{\mathbf{x}}\mathbb{S}(\kappa)$ | Tangent space attached with $\mathbf{x}$ on Riemannian manifold, Hyperbolic space, Hyperspherical space |
| $\oplus_{\kappa}$ | Addition operation of hyperbolic space |
| $d_{\kappa}(\cdot, \cdot)$ | Geodesic distance on hyperbolic/hyperspherical space |
| $\langle \cdot, \cdot \rangle$ | Inner product in vector space |
| $p_{\theta}(\cdot)$ | Generative model with parameter $\theta$ |
| $G = (V, E)$ | Graph with vertex set $V$ and edge set $E$ |
| $a_t$ | Action taken in time step $t$ |
| $\tau$ | Trajectory taken by the model |
| $T$ | Number of time steps |
| $D$ | The dimension of original feature space |
| $C$ | The number of mixed-curvature subspaces |
| $U_1, U_2$ | Metric spaces |
| $Exp_{\mathbf{x}}^{\kappa}(\cdot)$ | Exponential map attached with point $\mathbf{x}$ on the Riemannian manifold with curvature $\kappa$ |
| $Log_{\mathbf{x}}^{\kappa}(\cdot)$ | Logarithmic map attached with point $\mathbf{x}$ on the Riemannian manifold with curvature $\kappa$ |
| $\alpha, \beta$ | Learnable factors for balancing geometric information between layers |

*Table 6.* List of notations that appear in the paper.

| Type | Model | n=50 Gap | n=100 Gap | Type | Model | n=50 Gap | n=100 Gap |
|---|---|---|---|---|---|---|---|
| CVRP | POMO-MTL | 0.987% | 1.846% | VRPTW | POMO-MTL | 3.637% | 5.313% |
| | Mixed-POMO-MTL | **0.980%** | **1.731%** | | Mixed-POMO-MTL | **3.556%** | **5.090%** |
| | Euc-POMO-MTL | 1.033% | 1.815% | | Euc-POMO-MTL | 3.719% | 5.305% |
| | MVMoE-L | 0.955% | 1.728% | | MVMoE-L | 3.500% | 4.927% |
| | Mixed-MVMoE-L | **0.933%** | **1.645%** | | Mixed-MVMoE-L | **3.421%** | **4.872%** |
| | Euc-MVMoE-L | 0.965% | 1.743% | | Euc-MVMoE-L | 3.508% | 4.995% |
| | MVMoE | 0.896% | 1.653% | | MVMoE | 3.410% | 4.903% |
| | Mixed-MVMoE | **0.865%** | **1.599%** | | Mixed-MVMoE | **3.373%** | **4.732%** |
| | Euc-MVMoE | 0.900% | 1.672% | | Euc-MVMoE | 3.414% | 4.892% |
| OVRP | POMO-MTL | 1.684% | 1.674% | VRPL | POMO-MTL | 3.470% | 4.411% |
| | Mixed-POMO-MTL | **1.645%** | **1.427%** | | Mixed-POMO-MTL | 3.420% | 3.996% |
| | Euc-POMO-MTL | 1.713% | 1.569% | | Euc-POMO-MTL | 3.577% | 4.401% |
| | MVMoE-L | 1.605% | 1.368% | | MVMoE-L | 3.322% | 3.941% |
| | Mixed-MVMoE-L | **1.531%** | **1.265%** | | Mixed-MVMoE-L | **3.219%** | **3.749%** |
| | Euc-MVMoE-L | 1.602% | 1.427% | | Euc-MVMoE-L | 3.366% | 3.992% |
| | MVMoE | 1.540% | 1.285% | | MVMoE | 3.210% | 3.852% |
| | Mixed-MVMoE | **1.456%** | **1.153%** | | Mixed-MVMoE | **3.060%** | **3.579%** |
| | Euc-MVMoE | 1.535% | 1.304% | | Euc-MVMoE | 3.202% | 3.708% |
| VRPB | POMO-MTL | **2.634%** | 3.458% | OVRPTW | POMO-MTL | 0.201% | 0.479% |
| | Mixed-POMO-MTL | 2.637% | **3.312%** | | Mixed-POMO-MTL | **0.185%** | **0.362%** |
| | Euc-POMO-MTL | 2.656% | 3.479% | | Euc-POMO-MTL | 0.242% | 0.472% |
| | MVMoE-L | 2.548% | 3.214% | | MVMoE-L | 0.131% | 0.323% |
| | Mixed-MVMoE-L | **2.448%** | **3.133%** | | Mixed-MVMoE-L | **0.098%** | 0.270% |
| | Euc-MVMoE-L | 2.532% | 3.410% | | Euc-MVMoE-L | 0.365% | **0.096%** |
| | MVMoE | 2.402% | 3.136% | | MVMoE | 0.092% | 0.261% |
| | Mixed-MVMoE | **2.336%** | **2.946%** | | Mixed-MVMoE | **0.052%** | **0.227%** |
| | Euc-MVMoE | 2.410% | 3.157% | | Euc-MVMoE | 0.096% | 0.277% |
| OVRPB | POMO-MTL | 6.430% | **7.335%** | OVRPL | POMO-MTL | 2.734% | 3.441% |
| | Mixed-POMO-MTL | **6.348%** | 7.831% | | Mixed-POMO-MTL | 2.708% | **3.350%** |
| | Euc-POMO-MTL | 6.378% | 7.971% | | Euc-POMO-MTL | 2.762% | 3.453% |
| | MVMoE-L | 7.243% | 7.243% | | MVMoE-L | 2.597% | 3.244% |
| | Mixed-MVMoE-L | **6.175%** | **6.997%** | | Mixed-MVMoE-L | **2.497%** | 3.159% |
| | Euc-MVMoE-L | 6.525% | 7.580% | | Euc-MVMoE-L | 2.606% | **3.120%** |
| | MVMoE | 5.999% | 7.088% | | MVMoE | 2.454% | 3.148% |
| | Mixed-MVMoE | **5.871%** | **6.800%** | | Mixed-MVMoE | **2.419%** | **2.971%** |
| | Euc-MVMoE | 5.994% | 6.909% | | Euc-MVMoE | 2.459% | 3.141% |
| VRPBL | POMO-MTL | 1.971% | 1.793% | VRPBTW | POMO-MTL | **8.841%** | **7.413%** |
| | Mixed-POMO-MTL | **1.905%** | **1.514%** | | Mixed-POMO-MTL | 8.934% | 7.457% |
| | Euc-POMO-MTL | 1.969% | 1.693% | | Euc-POMO-MTL | 9.188% | 7.414% |
| | MVMoE-L | 1.872% | 1.473% | | MVMoE-L | 8.745% | 7.190% |
| | Mixed-MVMoE-L | **1.781%** | **1.378%** | | Mixed-MVMoE-L | **8.715%** | **7.018%** |
| | Euc-MVMoE-L | 1.886% | 1.516% | | Euc-MVMoE-L | 8.803% | 7.183% |
| | MVMoE | 1.776% | 1.346% | | MVMoE | 8.600% | 4.903% |
| | Mixed-MVMoE | **1.729%** | **1.264%** | | Mixed-MVMoE | **8.545%** | **4.732%** |
| | Euc-MVMoE | 1.779% | 1.405% | | Euc-MVMoE | 8.665% | 7.113% |
| VRPLTW | POMO-MTL | **2.586%** | 1.920% | OVRPBL | POMO-MTL | 6.306% | **7.343%** |
| | Mixed-POMO-MTL | 2.621% | **1.673%** | | Mixed-POMO-MTL | **6.282%** | 7.919% |
| | Euc-POMO-MTL | 2.720% | 1.926% | | Euc-POMO-MTL | 6.305% | 8.015% |
| | MVMoE-L | 2.535% | 1.545% | | MVMoE-L | 6.310% | 7.300% |
| | Mixed-MVMoE-L | **2.448%** | **1.515%** | | Mixed-MVMoE-L | **6.077%** | **7.027%** |
| | Euc-MVMoE-L | 2.530% | 1.618% | | Euc-MVMoE-L | 6.311% | 7.560% |
| | MVMoE | 2.421% | 1.471% | | MVMoE | 5.843% | 7.115% |
| | Mixed-MVMoE | **2.387%** | **1.365%** | | Mixed-MVMoE | **5.705%** | **6.857%** |

*Continued on the next page*

| Type | Model | n=50 Gap | n=100 Gap | Type | Model | n=50 Gap | n=100 Gap |
|---|---|---|---|---|---|---|---|
| | Euc-MVMoE | 2.418% | 1.492% | | Euc-MVMoE | 5.831% | 7.047% |
| OVRPBTW | POMO-MTL | **9.628%** | 10.453% | OVRPLTW | POMO-MTL | 3.633% | 4.374% |
| | Mixed-POMO-MTL | 9.734% | **10.192%** | | Mixed-POMO-MTL | **3.600%** | **4.020%** |
| | Euc-POMO-MTL | 9.785% | 10.666% | | Euc-POMO-MTL | 3.710% | 4.375% |
| | MVMoE-L | 9.630% | 10.188% | | MVMoE-L | 3.488% | 3.396% |
| | Mixed-MVMoE-L | **9.530%** | **9.899%** | | Mixed-MVMoE-L | **3.335%** | **3.816%** |
| | Euc-MVMoE-L | 9.592% | 10.163% | | Euc-MVMoE-L | 3.529% | 4.063% |
| | MVMoE | 9.308% | 9.948% | | MVMoE | 3.396% | 3.903% |
| | Mixed-MVMoE | **9.283%** | **9.749%** | | Mixed-MVMoE | **3.225%** | **3.560%** |
| | Euc-MVMoE | 9.443% | 9.968% | | Euc-MVMoE | 3.386% | 3.915% |
| VRPBLTW | POMO-MTL | **9.035%** | 7.746% | OVRPBLTW | POMO-MTL | **9.851%** | 10.498% |
| | Mixed-POMO-MTL | 9.139% | **7.658%** | | Mixed-POMO-MTL | 9.946% | **10.358%** |
| | Euc-POMO-MTL | 9.286% | 7.761% | | Euc-POMO-MTL | 9.967% | 10.670% |
| | MVMoE-L | 8.915% | 7.473% | | MVMoE-L | 9.682% | 10.263% |
| | Mixed-MVMoE-L | **8.871%** | **7.278%** | | Mixed-MVMoE-L | **9.582%** | **10.027%** |
| | Euc-MVMoE-L | 8.996% | 7.537% | | Euc-MVMoE-L | 9.754% | 10.247% |
| | MVMoE | 8.775% | 7.332% | | MVMoE | 9.516% | 10.009% |
| | Mixed-MVMoE | **8.690%** | **7.304%** | | Mixed-MVMoE | **9.462%** | **9.772%** |
| | Euc-MVMoE | 8.875% | 7.422% | | Euc-MVMoE | 9.636% | 10.045% |

Table 7: Ablation studies on whether the improvements on performances stem from the increased number of parameters or the design of mixed-curvature geometric spaces. The training configurations are consistent with (Zhou et al., 2024). Comparisons are conducted on MVMoE with 16 VRP variants (6 in-distribution and 10 out-of-distribution tasks) in which case each task contains 1,000 instances. Bold indicates best and underline indicates the second-best result. Euc-X represents the model that replaces the mixed-curvature modules with their Euclidean counterparts so that numbers of total parameters for these models are in the same level.

| Type | Model | n=50 Gap | n=100 Gap | Type | Model | n=50 Gap | n=100 Gap |
|---|---|---|---|---|---|---|---|
| CVRP | POMO-MTL | 0.987% | 1.846% | VRPTW | POMO-MTL | 3.637% | 5.313% |
| | Mixed-POMO-MTL | **0.980%** | **1.731%** | | Mixed-POMO-MTL | **3.556%** | **5.090%** |
| | Mixed-POMO-MTL (w.o. Mix-up) | 1.057% | 1.807% | | Mixed-POMO-MTL (w.o. Mix-up) | 3.647% | 5.156% |
| | MVMoE-L | 0.955% | 1.728% | | MVMoE-L | 3.500% | 4.927% |
| | Mixed-MVMoE-L | **0.933%** | **1.645%** | | Mixed-MVMoE-L | **3.421%** | **4.872%** |
| | Mixed-MVMoE-L (w.o. Mix-up) | 0.984% | 1.731% | | Mixed-MVMoE-L (w.o. Mix-up) | 3.516% | 5.018% |
| | MVMoE | 0.896% | 1.653% | | MVMoE | 3.410% | 4.903% |
| | Mixed-MVMoE | **0.865%** | **1.599%** | | Mixed-MVMoE | **3.373%** | **4.732%** |
| | Mixed-MVMoE (w.o. Mix-up) | 0.890% | 1.674% | | Mixed-MVMoE (w.o. Mix-up) | 3.256% | 4.897% |
| OVRP | POMO-MTL | 2.634% | 3.458% | VRPL | POMO-MTL | 0.201% | 0.479% |
| | Mixed-POMO-MTL | 2.637% | **3.312%** | | Mixed-POMO-MTL | **0.185%** | **0.362%** |
| | Mixed-POMO-MTL (w.o. Mix-up) | **2.607%** | 3.373% | | Mixed-POMO-MTL (w.o. Mix-up) | 0.205% | 0.420% |
| | MVMoE-L | 2.548% | 3.214% | | MVMoE-L | 0.131% | 0.323% |
| | Mixed-MVMoE-L | **2.448%** | **3.133%** | | Mixed-MVMoE-L | **0.098%** | **0.270%** |
| | Mixed-MVMoE-L (w.o. Mix-up) | 2.608% | 3.197% | | Mixed-MVMoE-L (w.o. Mix-up) | 0.150% | 0.350% |
| | MVMoE | 2.402% | 3.136% | | MVMoE | 0.092% | 0.261% |
| | Mixed-MVMoE | **2.336%** | **2.946%** | | Mixed-MVMoE | **0.052%** | **0.227%** |
| | Mixed-MVMoE (w.o. Mix-up) | 2.421% | 3.129% | | Mixed-MVMoE (w.o. Mix-up) | 0.086% | 0.294% |
| VRPB | POMO-MTL | 1.684% | 1.674% | OVRPTW | POMO-MTL | 3.470% | 4.411% |
| | Mixed-POMO-MTL | **1.645%** | **1.427%** | | Mixed-POMO-MTL | 3.420% | **3.996%** |
| | Mixed-POMO-MTL (w.o. Mix-up) | 1.682% | 1.517% | | Mixed-POMO-MTL (w.o. Mix-up) | 3.599% | 4.136% |
| | MVMoE-L | 1.605% | 1.368% | | MVMoE-L | 3.322% | 3.941% |
| | Mixed-MVMoE-L | **1.531%** | **1.265%** | | Mixed-MVMoE-L | **3.219%** | **3.749%** |
| | Mixed-MVMoE-L (w.o. Mix-up) | 1.591% | 1.418% | | Mixed-MVMoE-L (w.o. Mix-up) | 3.431% | 4.001% |
| | MVMoE | 1.540% | 1.285% | | MVMoE | 3.210% | 3.852% |
| | Mixed-MVMoE | **1.456%** | **1.153%** | | Mixed-MVMoE | **3.060%** | **3.579%** |
| | Mixed-MVMoE (w.o. Mix-up) | 1.540% | 1.307% | | Mixed-MVMoE (w.o. Mix-up) | 3.256% | 3.939% |
| OVRPB | POMO-MTL | 6.430% | **7.335%** | OVRPL | POMO-MTL | 2.734% | 3.441% |
| | Mixed-POMO-MTL | **6.348%** | 7.831% | | Mixed-POMO-MTL | **2.708%** | 3.350% |
| | Mixed-POMO-MTL (w.o. Mix-up) | 6.298% | 7.414% | | Mixed-POMO-MTL (w.o. Mix-up) | 2.830% | **3.332%** |
| | MVMoE-L | 7.243% | 7.243% | | MVMoE-L | 2.597% | 3.244% |
| | Mixed-MVMoE-L | **6.175%** | **6.997%** | | Mixed-MVMoE-L | **2.497%** | **3.159%** |
| | Mixed-MVMoE-L (w.o. Mix-up) | 6.241% | 7.094% | | Mixed-MVMoE-L (w.o. Mix-up) | 2.668% | 3.173% |
| | MVMoE | 5.999% | 7.088% | | MVMoE | 2.454% | 3.148% |
| | Mixed-MVMoE | **5.871%** | **6.800%** | | Mixed-MVMoE | **2.419%** | **2.971%** |
| | Mixed-MVMoE (w.o. Mix-up) | 5.760% | 6.782% | | Mixed-MVMoE (w.o. Mix-up) | 2.505% | 3.152% |
| VRPBL | POMO-MTL | 1.971% | 1.793% | VRPBTW | POMO-MTL | **8.841%** | 7.413% |
| | Mixed-POMO-MTL | **1.905%** | **1.514%** | | Mixed-POMO-MTL | 8.934% | 7.457% |
| | Mixed-POMO-MTL (w.o. Mix-up) | 1.987% | 1.639% | | Mixed-POMO-MTL (w.o. Mix-up) | 8.900% | **7.383%** |
| | MVMoE-L | 1.872% | 1.473% | | MVMoE-L | 8.745% | 7.190% |
| | Mixed-MVMoE-L | **1.781%** | **1.378%** | | Mixed-MVMoE-L | **8.715%** | **7.018%** |
| | Mixed-MVMoE-L (w.o. Mix-up) | 1.860% | 1.531% | | Mixed-MVMoE-L (w.o. Mix-up) | 8.790% | 7.169% |
| | MVMoE | 1.776% | 1.346% | | MVMoE | 8.600% | 4.903% |
| | Mixed-MVMoE | **1.729%** | **1.264%** | | Mixed-MVMoE | **8.545%** | **4.732%** |
| | Mixed-MVMoE (w.o. Mix-up) | 1.773% | 1.430% | | Mixed-MVMoE (w.o. Mix-up) | 8.649% | 7.041% |
| VRPLTW | POMO-MTL | **2.586%** | 1.920% | OVRPBL | POMO-MTL | 6.306% | **7.343%** |
| | Mixed-POMO-MTL | 2.621% | **1.673%** | | Mixed-POMO-MTL | 6.282% | 7.919% |
| | Mixed-POMO-MTL (w.o. Mix-up) | 2.670% | 1.805% | | Mixed-POMO-MTL (w.o. Mix-up) | **6.111%** | 7.460% |
| | MVMoE-L | 2.535% | 1.545% | | MVMoE-L | 6.310% | 7.300% |
| | Mixed-MVMoE-L | **2.448%** | 1.515% | | Mixed-MVMoE-L | **6.077%** | **7.027%** |
| | Mixed-MVMoE-L (w.o. Mix-up) | 2.533% | 1.601% | | Mixed-MVMoE-L (w.o. Mix-up) | 6.198% | 7.109% |
| | MVMoE | 2.421% | 1.471% | | MVMoE | 5.843% | 7.115% |
| | Mixed-MVMoE | **2.387%** | **1.365%** | | Mixed-MVMoE | **5.705%** | **6.857%** |

*Continued on the next page*

| Type | Model | n=50 Gap | n=100 Gap | Type | Model | n=50 Gap | n=100 Gap |
|---|---|---|---|---|---|---|---|
| | Mixed-MVMoE (w.o. Mix-up) | 2.478% | 1.523% | | Mixed-MVMoE (w.o. Mix-up) | 5.705% | 6.809% |
| OVRPBTW | POMO-MTL | **9.628%** | 10.453% | OVRPLTW | POMO-MTL | 3.633% | 4.374% |
| | Mixed-POMO-MTL | 9.734% | **10.192%** | | Mixed-POMO-MTL | **3.600%** | **4.020%** |
| | Mixed-POMO-MTL (w.o. Mix-up) | 9.818% | 10.251% | | Mixed-POMO-MTL (w.o. Mix-up) | 3.765% | 4.124% |
| | MVMoE-L | 9.630% | 10.188% | | MVMoE-L | 3.488% | 3.396% |
| | Mixed-MVMoE-L | **9.530%** | **9.899%** | | Mixed-MVMoE-L | **3.335%** | **3.816%** |
| | Mixed-MVMoE-L (w.o. Mix-up) | 9.639% | 10.032% | | Mixed-MVMoE-L (w.o. Mix-up) | 3.546% | 4.037% |
| | MVMoE | 9.308% | 9.948% | | MVMoE | 3.396% | 3.903% |
| | Mixed-MVMoE | **9.283%** | **9.749%** | | Mixed-MVMoE | **3.225%** | **3.560%** |
| | Mixed-MVMoE (w.o. Mix-up) | 9.441% | 10.096% | | Mixed-MVMoE (w.o. Mix-up) | 3.434% | 3.932% |
| VRPBLTW | POMO-MTL | **9.035%** | 7.746% | OVRPBLTW | POMO-MTL | **9.851%** | 10.498% |
| | Mixed-POMO-MTL | 9.139% | **7.658%** | | Mixed-POMO-MTL | 9.946% | **10.358%** |
| | Mixed-POMO-MTL (w.o. Mix-up) | 9.102% | 7.699% | | Mixed-POMO-MTL (w.o. Mix-up) | 9.940% | 10.323% |
| | MVMoE-L | 8.915% | 7.473% | | MVMoE-L | 9.682% | 10.263% |
| | Mixed-MVMoE-L | **8.871%** | **7.278%** | | Mixed-MVMoE-L | **9.582%** | **10.027%** |
| | Mixed-MVMoE-L (w.o. Mix-up) | 9.013% | 7.410% | | Mixed-MVMoE-L (w.o. Mix-up) | 9.764% | 10.135% |
| | MVMoE | 8.775% | 7.332% | | MVMoE | 9.516% | 10.009% |
| | Mixed-MVMoE | **8.690%** | **7.304%** | | Mixed-MVMoE | **9.462%** | **9.772%** |
| | Mixed-MVMoE (w.o. Mix-up) | 8.881% | 7.351% | | Mixed-MVMoE (w.o. Mix-up) | 9.579% | 10.201% |

Table 8: Ablation studies on whether the Mix-up module brings improvements on performances. The training configurations are consistent with (Zhou et al., 2024). Comparisons are conducted on MVMoE with 16 VRP variants (6 in-distribution and 10 out-of-distribution tasks) in which case each task contains 1,000 instances. Bold indicates best and underline indicates the second-best result.

| Model | Parameters |
|---|---|
| POMO-MTL | 1,254,656 |
| Mixed-POMO-MTL | 1,386,810 |
| MVMoE-Light | 3,698,944 |
| Mixed-MVMoE-Light | 3,831,116 |
| MVMoE | 3,682,176 |
| Mixed-MVMoE | 3,814,348 |

Table 9. Comparisons for the number of parameters in each baseline model and their mixed-curvature space counterparts. Our comparisons are based on the models mentioned in (Zhou et al., 2024). For the embedder, we insert two mixed-curvature modules for processing features from depot and customer nodes, respectively. For each layer of encoder, we insert one mixed-curvature module for processing features from the previous layer. In specifics, the ratio of increased parameters is between 3.57% and 10.56% compared to baselines.

| Type | Model | n=50 Gap | n=100 Gap | Type | Model | n=50 Gap | n=100 Gap |
|---|---|---|---|---|---|---|---|
| CVRP | Mixed-POMO-MTL-4 | 1.011% | 1.840% | VRPTW | Mixed-POMO-MTL-4 | 3.583% | 5.205% |
| | Mixed-POMO-MTL-8 | **0.980%** | **1.731%** | | Mixed-POMO-MTL-8 | **3.556%** | **5.090%** |
| | Mixed-POMO-MTL-16 | 1.009% | 1.808% | | Mixed-POMO-MTL-16 | 3.570% | 5.250% |
| OVRP | Mixed-POMO-MTL-4 | 2.645% | 3.465% | VRPL | Mixed-POMO-MTL-4 | 0.193% | 0.450% |
| | Mixed-POMO-MTL-8 | 2.637% | **3.312%** | | Mixed-POMO-MTL-8 | **0.185%** | **0.362%** |
| | Mixed-POMO-MTL-16 | **2.592%** | 3.367% | | Mixed-POMO-MTL-16 | 0.187% | 0.456% |
| VRPB | Mixed-POMO-MTL-4 | 1.678% | 1.585% | OVRPTW | Mixed-POMO-MTL-4 | 3.423% | 4.361% |
| | Mixed-POMO-MTL-8 | **1.645%** | **1.427%** | | Mixed-POMO-MTL-8 | **3.420%** | **3.996%** |
| | Mixed-POMO-MTL-16 | 1.655% | 1.564% | | Mixed-POMO-MTL-16 | 3.475% | 4.234% |
| OVRPB | Mixed-POMO-MTL-4 | **6.279%** | 7.470% | OVRPL | Mixed-POMO-MTL-4 | 2.737% | **2.692%** |
| | Mixed-POMO-MTL-8 | 6.348% | 7.831% | | Mixed-POMO-MTL-8 | **2.708%** | 3.350% |
| | Mixed-POMO-MTL-16 | 6.290% | **7.328%** | | Mixed-POMO-MTL-16 | 3.356% | 3.349% |
| VRPBL | Mixed-POMO-MTL-4 | 1.947% | 1.729% | VRPBTW | Mixed-POMO-MTL-4 | **8.820%** | 7.548% |
| | Mixed-POMO-MTL-8 | **1.905%** | **1.514%** | | Mixed-POMO-MTL-8 | 8.934% | 7.457% |
| | Mixed-POMO-MTL-16 | 1.956% | 1.696% | | Mixed-POMO-MTL-16 | 8.847% | **7.402%** |
| VRPLTW | Mixed-POMO-MTL-4 | **2.520%** | 1.830% | OVRPBL | Mixed-POMO-MTL-4 | **6.151%** | 7.444% |
| | Mixed-POMO-MTL-8 | 2.621% | **1.673%** | | Mixed-POMO-MTL-8 | 6.282% | 7.919% |
| | Mixed-POMO-MTL-16 | 2.585% | 1.891% | | Mixed-POMO-MTL-16 | 6.160% | **7.376%** |
| OVRPBTW | Mixed-POMO-MTL-4 | 9.788% | 10.470% | OVRPLTW | Mixed-POMO-MTL-4 | 3.630% | 4.382% |
| | Mixed-POMO-MTL-8 | 9.734% | **10.192%** | | Mixed-POMO-MTL-8 | 3.600% | **4.020%** |
| | Mixed-POMO-MTL-16 | 9.698% | 10.415% | | Mixed-POMO-MTL-16 | **3.573%** | 4.270% |
| VRPBLTW | Mixed-POMO-MTL-4 | **9.022%** | 7.731% | OVRPBLTW | Mixed-POMO-MTL-4 | **9.836%** | 10.583% |
| | Mixed-POMO-MTL-8 | 9.139% | **7.658%** | | Mixed-POMO-MTL-8 | 9.946% | **10.358%** |
| | Mixed-POMO-MTL-16 | 9.112% | 7.733% | | Mixed-POMO-MTL-16 | 9.868% | 10.488% |

*Table 10.* Ablation studies on the number of subspaces in mixed-curvature modules. The training configurations are consistent with (Zhou et al., 2024). Comparisons are conducted on POMO-MTL with 16 VRP variants in which cases each task contains 1,000 instances. Bold indicates best value and the underline indicates the second best result.

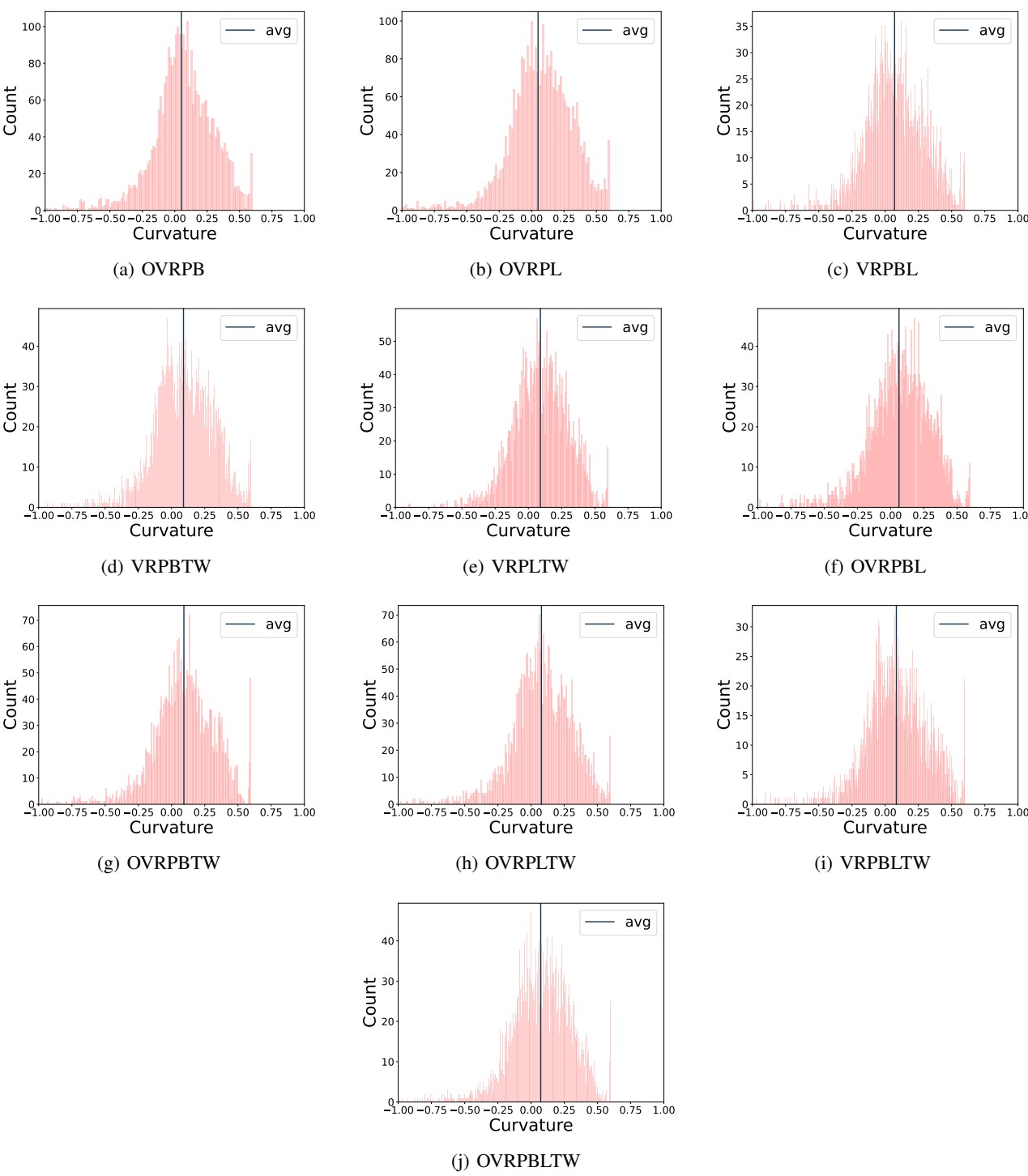

(a) OVRPB     (b) OVRPL     (c) VRPBL

(d) VRPBTW     (e) VRPLTW     (f) OVRPBL

(g) OVRPBTW     (h) OVRPLTW     (i) VRPBLTW

(j) OVRPBLTW

*Figure 4.* The histogram of curvatures on each node from the remaining 10 VRP tasks. We utilize 1,000 instances with size 50 for each task to visualize curvature information. The x-axis represents curvature values, while the y-axis denotes the count of each value. The avg line indicates the average curvature across all nodes. We adopt Ollivier-Ricci curvature (Ollivier, 2009) which is especially suitable for measuring curvatures on discrete structures like graphs. From the information in the figure, it shows that almost every node in each task dataset has either negative or positive curvature and the average curvature suggests that each task contains non-Euclidean geometry patterns. Better viewed in color.

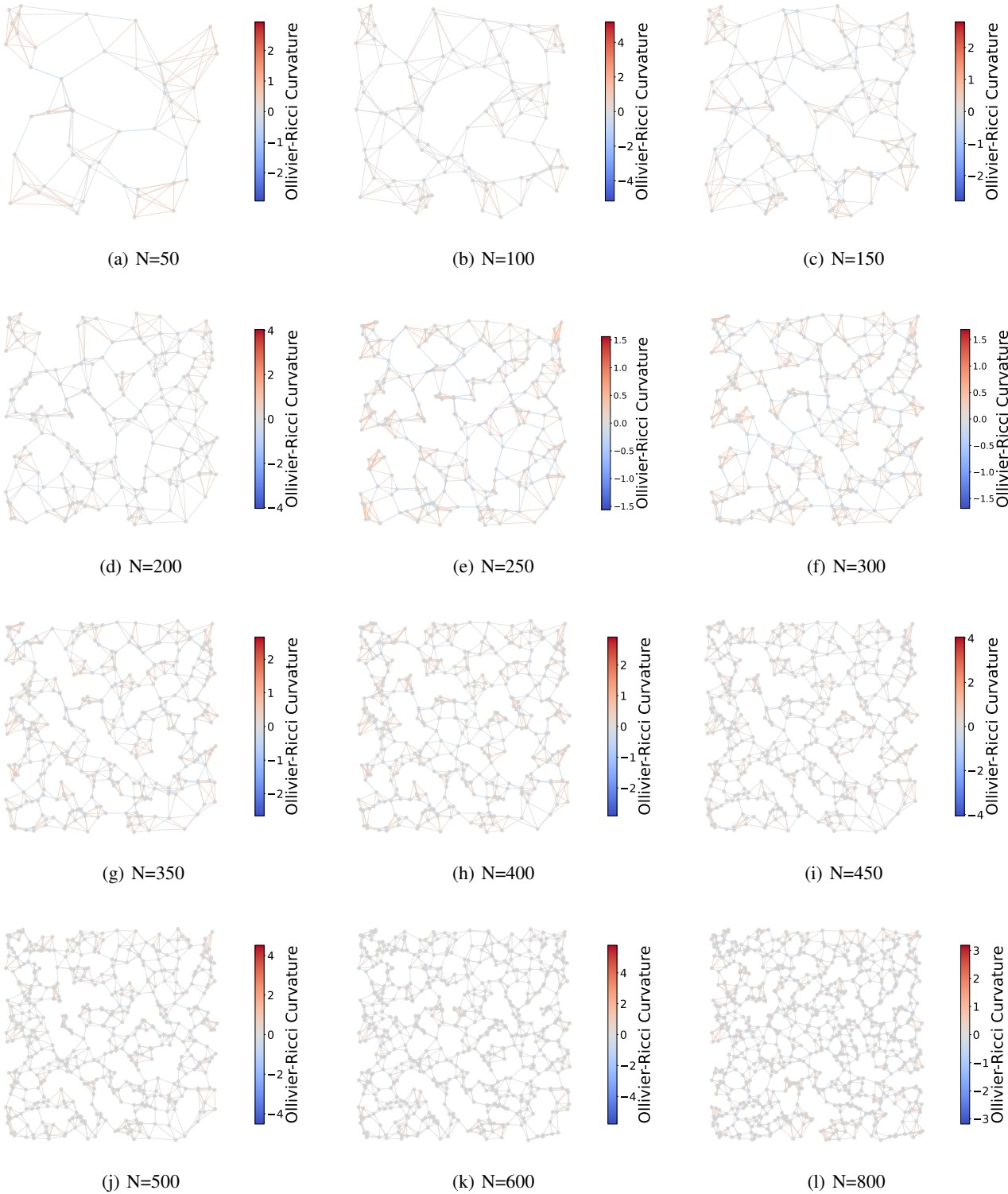

*Figure 5.* Visualization of Ollivier-Ricci curvatures on random generated complete graphs with different sizes. Each node's coordinate is restricted in the 2D region $[-1, 1] \times [-1, 1]$. For each complete graph, we use KNN to select 5 nearest neighbors of each node. Warm (cold) colors represent positive (negative) curvatures. From these presented results, we can observe that edges with negative curvatures often link distant nodes while edges with positive curvatures often exist in highly clustered regions. This property is shared among all of the node sizes. Better viewed in color.

**Appendix.2. Detailed Experimental Configurations with (Zhou et al., 2024)**

**Definitions of Constraints in Utilized Tasks**

We follow the settings of (Kool et al., 2019; Kwon et al., 2020; Zhou et al., 2024) and details are listed as follows:

- **Coordinates:** We focus on uniform distribution setting in which case each node's locations are sampled from $U(0, 1)$ in a unit square.

- **Capacity:** We set capacity to 40 and 50 for $N = 50$ and $N = 100$, respectively. Note that one of the hard constraints involved in each task is that nodes with demands greater than delivery vehicle's current demand are masked.

- **Demand:** We sample the demand of each node from the list $\{1, 2, \ldots, 9\}$. Note that before sending into model, the node demand is normalized by the demand of delivery vehicle.

- **Open Route:** We set it as an indicator vector with all ones. During decoding stage, we need to manually set mask to prevent delivery vehicle from going back to the depot node.

- **Backhauls:** Similar to demand setting, we sample from $\{1, 2, \ldots, 9\}$ as our initial demands. Then, in the same way as that of (Liu et al., 2024), we sample 20% of nodes to be the backhauls nodes.

- **Duration Limit:** We set it to 3, which represents the maximum length of delivery vehicle's route.

- **Time Window:** For the depot node, we assign its time window as $[0, 3]$ and service time for depot is 0 by default. However, service time for customer nodes is set to 0.2 and time window for customer nodes are sampled from uniform distribution.

By combining constraints in different ways, we can obtain various kinds of tasks as listed in Table 11. Since some nodes don't have features like time-windows or backhauls, these features will separately appear in the encoder module.

|  | Capacity (C) | Open Route (O) | Backhauls (B) | Duration Limit (L) | Time Window (TW) |
|---|---|---|---|---|---|
| CVRP | ✓ | ✗ | ✗ | ✗ | ✗ |
| VRPTW | ✓ | ✗ | ✗ | ✗ | ✓ |
| OVRP | ✓ | ✓ | ✗ | ✗ | ✗ |
| VRPL | ✓ | ✗ | ✗ | ✓ | ✗ |
| VRPB | ✓ | ✗ | ✓ | ✗ | ✗ |
| OVRPTW | ✓ | ✓ | ✗ | ✗ | ✓ |
| OVRPB | ✓ | ✓ | ✓ | ✗ | ✗ |
| OVRPL | ✓ | ✓ | ✗ | ✓ | ✗ |
| VRPBL | ✓ | ✗ | ✓ | ✓ | ✗ |
| VRPBTW | ✓ | ✗ | ✓ | ✗ | ✓ |
| VRPLTW | ✓ | ✗ | ✗ | ✓ | ✓ |
| OVRPBL | ✓ | ✓ | ✓ | ✓ | ✗ |
| OVRPBTW | ✓ | ✓ | ✓ | ✗ | ✓ |
| OVRPLTW | ✓ | ✓ | ✗ | ✓ | ✓ |
| VRPBLTW | ✓ | ✗ | ✓ | ✓ | ✓ |
| OVRPBLTW | ✓ | ✓ | ✓ | ✓ | ✓ |

*Table 11.* Detailed descriptions of constraints contained in each problem type. We have 16 VRP tasks in total. The first 6 VRP tasks get involved in training stage and the last 10 tasks are used for zero-shot/few-shot testings.

| Hyper-Parameters | Value |
|---|---|
| Training Epochs | 5,000 |
| Fine-tuning Epochs | 10 |
| Instances in each Training Epoch | 20,000 |
| Instances in each Fine-tuning Epoch | 10,000 |
| Optimizer | Adam |
| LR Scheduler | MultiStepLR |
| LR Milestones | [4,501] |
| LR Gamma | 0.1 |
| Training Learning Rate | 1e-4 |
| Fine-tuning Learning Rate | 1e-4 |
| Weight Decay | 1e-6 |
| Training Batch Size | 128 |
| Fine-tuning Batch Szie | 128 |
| Evaluation Batch Size | 64 |
| Problem Scales | $\{50, 100\}$ |
| Node Distribution | $U(0,1)$ |
| Number of Experts in MoE | 4 |
| Auxiliary Loss Weight in MoE | 0.001 |
| Gating Mechanism in MoE | node-level, input-choice gating |
| Embedding Size | 128 |
| Hidden Feature Size | 512 |
| Number of Encoder Layers | 6 |
| QKV Dimension | 16 |
| Attention Head Number | 8 |
| Logit Clipping | 10 |
| Evaluation Type | argmax |
| Number of Experts in MoE for Routing | 2 |
| Number of Subspaces ($C$) | 8 |
| Initialization value of Curvature ($\kappa$) | 0 |
| Initialization value of $\alpha, \beta$ | $\{1,1\}$ |

*Table 12.* Detailed experiment settings of hyper-parameters. This configuration is consistent with (Zhou et al., 2024). However, other choices for the number of subspaces are also valid as long as the sum of subspaces' dimensions equals 128. Even if 128 is not divisible by number of subspaces, we can still determine dimensions manually or automatically (*e.g.*, neural architecture search (Elsken et al., 2019)).

**Appendix.3. Real-World Experimental Results with (Zhou et al., 2024) and Subspace Visualizations**

| Set-Solomon | | POMO | | POMO-MTL | | MVMoE | | Mixed-POMO-MTL | |
|---|---|---|---|---|---|---|---|---|---|
| Instance | Opt | Obj | Gap | Obj | Gap | Obj | Gap | Obj | Gap |
| R101 | 1637.7 | **1805.6** | **10.252%** | 1821.2 | 11.205% | 1798.1 | 9.794% | 1862.3 | 13.714% |
| R102 | 1466.6 | **1556.7** | **6.143%** | 1596.0 | 8.823% | 1572.0 | 7.187% | 1634.1 | 11.422% |
| R103 | 1208.7 | 1341.4 | 10.979% | **1327.3** | **9.812%** | 1328.2 | 9.887% | 1374.1 | 13.687% |
| R104 | 971.5 | **1118.6** | **15.142%** | 1120.7 | 15.358% | 1124.8 | 15.780% | 1134.4 | 16.767% |
| R105 | 1355.3 | 1506.4 | 11.149% | 1514.6 | 11.754% | **1479.4** | **9.157%** | 1569.7 | 15.818% |
| R106 | 1234.6 | 1365.2 | 10.578% | 1380.5 | 11.818% | **1362.4** | **10.352%** | 1413.4 | 14.480% |
| R107 | 1064.6 | 1214.2 | 14.052% | 1209.3 | 13.592% | **1182.1** | **11.037%** | 1230.2 | 15.556% |
| R108 | 932.1 | 1058.9 | 13.604% | 1061.8 | 13.915% | **1023.2** | **9.774%** | 1063.0 | 14.046% |
| R109 | 1146.9 | **1249.0** | **8.902%** | 1265.7 | 10.358% | 1255.6 | 9.478% | 1258.2 | 9.704% |
| R110 | 1068.0 | **1180.4** | **10.524%** | 1171.4 | 9.682% | 1185.7 | 11.021% | 1213.2 | 13.593% |
| R111 | 1048.7 | 1177.2 | 12.253% | 1211.5 | 15.524% | **1176.1** | **12.148%** | 1189.8 | 13.453% |
| R112 | 948.6 | 1063.1 | 12.070% | 1057.0 | 11.427% | **1045.2** | **10.183%** | 1097.3 | 15.676% |
| RC101 | 1619.8 | 2643.0 | 63.168% | 1833.3 | 13.181% | **1774.4** | **9.544%** | 1882.7 | 16.231% |
| RC102 | 1457.4 | **1534.8** | **5.311%** | 1546.1 | 6.086% | 1544.5 | 5.976% | 1616.4 | 10.907% |
| RC103 | 1258.0 | 1407.5 | 11.884% | **1396.2** | **10.986%** | 1402.5 | 11.486% | 1403.0 | 11.526% |
| RC104 | 1132.3 | 1261.8 | 11.437% | 1271.7 | 12.311% | 1265.4 | 11.755% | **1252.6** | **10.628%** |
| RC105 | 1513.7 | **1612.9** | **6.553%** | 1644.9 | 8.668% | 1635.5 | 8.047% | 1660.1 | 8.382% |
| RC106 | 1372.7 | 1539.3 | 12.137% | 1552.8 | 13.120% | 1505.0 | 9.638% | **1497.2** | **9.072%** |
| RC107 | 1207.8 | 1347.7 | 11.583% | 1384.8 | 14.655% | 1351.6 | 11.906% | **1330.8** | **10.180%** |
| RC108 | 1114.2 | 1305.5 | 17.169% | 1274.4 | 14.378% | **1254.2** | **12.565%** | 1273.9 | 14.332% |
| RC201 | 1261.8 | 2045.6 | 62.118% | 1761.1 | 39.570% | **1577.3** | **25.004%** | 1595.3 | 26.428% |
| RC202 | 1092.3 | 1805.1 | 65.257% | 1486.2 | 36.062% | 1616.5 | 47.990% | **1416.4** | **29.672%** |
| RC203 | 923.7 | 1470.4 | 59.186% | 1360.4 | 47.277% | 1473.5 | 59.521% | **1223.3** | **32.433%** |
| RC204 | 783.5 | 1323.9 | 68.973% | 1331.7 | 69.968% | 1286.6 | 64.212% | **1103.8** | **40.887%** |
| RC205 | 1154.0 | 1568.4 | 35.910% | 1539.2 | 33.380% | 1537.7 | 33.250% | **1365.2** | **18.301%** |
| RC206 | 1051.1 | 1707.5 | 62.449% | 1472.6 | 40.101% | 1468.9 | 39.749% | **1239.7** | **17.939%** |
| RC207 | 962.9 | 1567.2 | 62.758% | 1375.7 | 42.870% | 1442.0 | 49.756% | **1264.7** | **31.345%** |
| RC208 | 776.1 | 1505.4 | 93.970% | 1185.6 | 52.764% | 1107.4 | 42.688% | **1113.0** | **43.407%** |
| Average Gap | | 29.658% | | 21.380% | | 20.317% | | **17.84%** | |

*Table 13.* Zero-Shot Inference on VRPTW benchmark instances from Set-Solomon. Each model is trained on the size n=100, following the settings in (Zhou et al., 2024).

| Set-X | | POMO | | POMO-MTL | | MVMoE | | Mixed-POMO-MTL | |
|---|---|---|---|---|---|---|---|---|---|
| Instance | Opt | Obj | Gap | Obj | Gap | Obj | Gap | Obj | Gap |
| X-n101-k25 | 27591 | 30138 | 9.231% | 32482 | 17.727% | **29361** | **6.415%** | 29676 | 7.557% |
| X-n106-k14 | 26362 | 39322 | 49.162% | 27369 | 3.820% | **27278** | **3.475%** | 27821 | 5.936% |
| X-n110-k13 | 14971 | 15223 | 1.683% | 15151 | 1.202% | **15089** | **0.788%** | 15226 | 1.703% |
| X-n115-k10 | 12747 | 16113 | 26.406% | 14785 | 15.988% | 13847 | 8.629% | **13328** | **4.558%** |
| X-n120-k6 | 13332 | 14085 | 5.648% | **13931** | **4.493%** | 14089 | 5.678% | 14039 | 5.303% |
| X-n125-k30 | 55539 | **58513** | **5.355%** | 60687 | 9.269% | 58944 | 6.131% | 59642 | 7.388% |
| X-n129-k18 | 28940 | **29246** | **1.057%** | 30332 | 4.810% | 29802 | 2.979% | 29476 | 1.852% |
| X-n134-k13 | 10916 | 11302 | 3.536% | 11581 | 6.092% | 11353 | 4.003% | **11298** | **3.499%** |
| X-n139-k10 | 13590 | 14035 | 3.274% | 13911 | 2.362% | 13825 | 1.729% | **13760** | **1.251%** |
| X-n143-k7 | 15700 | 16131 | 2.745% | 16660 | 6.115% | **16125** | **2.707%** | 16070 | 2.357% |
| X-n148-k46 | 43448 | 49328 | 13.533% | 50782 | 16.880% | **46758** | **7.618%** | 47157 | 8.537% |
| X-n153-k22 | 21220 | 32476 | 53.040% | 26237 | 23.643% | 23793 | 12.125% | **23392** | **10.236%** |
| X-n157-k13 | 16876 | 17660 | 4.646% | **17510** | **3.757%** | 17650 | 4.586% | 18444 | 9.291% |
| X-n162-k11 | 14138 | 14889 | 5.312% | 14720 | 4.117% | 14654 | 3.650% | **14588** | **3.183%** |
| X-n167-k10 | 20557 | 21822 | 6.154% | 21399 | 4.096% | 21340 | 3.809% | **21141** | **2.841%** |
| X-n172-k51 | 45607 | 49556 | 8.659% | 56385 | 23.632% | 51292 | 12.465% | **48815** | **7.034%** |
| X-n176-k26 | 47812 | 54197 | 13.354% | 57637 | 20.549% | 55520 | 16.121% | **52593** | **10.000%** |
| X-n181-k23 | 25569 | 37311 | 45.923% | **26219** | **2.542%** | 26258 | 2.695% | 27552 | 7.755% |
| X-n186-k15 | 24145 | 25222 | 4.461% | 25000 | 3.541% | 25182 | 4.295% | **24900** | **3.127%** |
| X-n190-k8 | 16980 | 18315 | 7.862% | **18113** | **6.673%** | 18327 | 7.933% | 18593 | 9.499% |
| X-n195-k51 | 44225 | 49158 | 11.154% | 54090 | 22.306% | 49984 | 13.022% | **48689** | **10.094%** |
| X-n200-k36 | 58578 | 64618 | 10.311% | 61654 | 5.251% | **61530** | **5.039%** | 61844 | 5.575% |
| X-n209-k16 | 30656 | 32212 | 5.076% | 32011 | 4.420% | 32033 | 4.492% | **31828** | **3.823%** |
| X-n219-k73 | 117595 | 133545 | 13.564% | **119887** | **1.949%** | 121046 | 2.935% | 125002 | 6.299% |
| X-n228-k23 | 25742 | 48689 | 89.142% | 33091 | 28.549% | 31054 | 20.636% | **29244** | **13.604%** |
| X-n237-k14 | 27042 | 29893 | 10.543% | **28472** | **5.288%** | 28550 | 5.577% | 28850 | 6.686% |
| X-n247-k50 | 37274 | 56167 | 50.687% | 45065 | 20.902% | 43673 | 17.167% | **41142** | **10.377%** |
| X-n251-k28 | 38684 | **40263** | **4.082%** | 40614 | 4.989% | 41022 | 6.044% | 40792 | 5.449% |
| Average Gap | | 16.629% | | 9.820% | | 6.884% | | **6.243%** | |

*Table 14.* Zero-Shot Inference on CVRP benchmark instances from Set-X. Each model is trained on the size n=100, following the settings in (Zhou et al., 2024).

| Set-X | | POMO | | POMO-MTL | | MVMoE | | Mixed-POMO-MTL | |
|---|---|---|---|---|---|---|---|---|---|
| Instance | Opt | Obj | Gap | Obj | Gap | Obj | Gap | Obj | Gap |
| X-n502-k39 | 69226 | 75617 | 9.232% | 77284 | 11.640% | **73533** | **6.222%** | 81423 | 17.619% |
| X-n513-k21 | 24201 | 30518 | 26.102% | **28510** | **17.805%** | 32102 | 32.647% | 29529 | 22.016% |
| X-n524-k153 | 154593 | 201877 | 30.586% | 192249 | 24.358% | 186540 | 20.665% | **173928** | **12.507%** |
| X-n536-k96 | 94846 | 106073 | 11.837% | 106514 | 12.302% | 109581 | 15.536% | **105632** | **11.372%** |
| X-n548-k50 | 86700 | 103093 | 18.908% | **94562** | **9.068%** | 95894 | 10.604% | 95680 | 10.358% |
| X-n561-k42 | 42717 | 49370 | 15.575% | **47846** | **12.007%** | 56008 | 31.114% | 49619 | 16.158% |
| X-n573-k30 | 50673 | 83545 | 64.871% | 60913 | 20.208% | 59473 | 17.366% | **57588** | **13.646%** |
| X-n586-k159 | 190316 | 229887 | 20.792% | **208893** | **9.761%** | 215668 | 13.321% | 212404 | 11.606% |
| X-n599-k92 | 108451 | 150572 | 38.839% | **120333** | **10.956%** | 128949 | 18.901% | 120722 | 11.315% |
| X-n613-k62 | 59535 | 68451 | 14.976% | **67984** | **14.192%** | 82586 | 38.718% | 71275 | 19.719% |
| X-n627-k43 | 62164 | 84434 | 35.825% | 73060 | 17.528% | 70987 | 14.193% | **69334** | **11.534%** |
| X-n641-k35 | 63682 | 75573 | 18.672% | 72643 | 14.071% | 75329 | 18.289% | **71750** | **12.669%** |
| X-n655-k131 | 106780 | 127211 | 19.134% | **116988** | **9.560%** | 117678 | 10.206% | 120227 | 12.593% |
| X-n670-k130 | 146332 | 208079 | 42.197% | 190118 | 29.922% | 197695 | 35.100% | **170403** | **16.450%** |
| X-n685-k75 | 68205 | **79482** | **16.534%** | 80892 | 18.601% | 97388 | 42.787% | 80512 | 18.044% |
| X-n701-k44 | 81923 | 97843 | 19.433% | 92075 | 12.392% | 98469 | 20.197% | **90724** | **10.734%** |
| X-n716-k35 | 43373 | 51381 | 18.463% | 52709 | 21.525% | 56773 | 30.895% | **50798** | **17.119%** |
| X-n733-k159 | 136187 | **159098** | **16.823%** | 161961 | 18.925% | 178322 | 30.939% | 161089 | 18.285% |
| X-n749-k98 | 77269 | **87786** | **13.611%** | 90582 | 17.229% | 100438 | 29.985% | 87907 | 13.767% |
| X-n766-k71 | 114417 | 135464 | 18.395% | 144041 | 25.891% | 152352 | 33.155% | **128375** | **12.199%** |
| X-n783-k48 | 72386 | 90289 | 24.733% | **83169** | **14.897%** | 100383 | 38.677% | 84181 | 16.295% |
| X-n801-k40 | 73305 | 124278 | 69.536% | **85077** | **16.059%** | 91560 | 24.903% | 86152 | 17.525% |
| X-n819-k171 | 158121 | 193451 | 22.344% | **177157** | **12.039%** | 183599 | 16.113% | 183792 | 16.235% |
| X-n837-k142 | 193737 | 237884 | 22.787% | 214207 | 10.566% | 229526 | 18.473% | **213651** | **10.279%** |
| X-n856-k95 | 88965 | 152528 | 71.447% | 101774 | 14.398% | **99129** | **11.425%** | 115359 | 29.668% |
| X-n876-k59 | 99299 | 119764 | 20.609% | 116617 | 17.440% | 119619 | 20.463% | **112067** | **12.858%** |
| X-n895-k37 | 53860 | 70245 | 30.421% | **65587** | **21.773%** | 79018 | 46.710% | 69614 | 29.250% |
| X-n916-k207 | 329179 | 399372 | 21.324% | **361719** | **9.885%** | 383681 | 16.557% | 365822 | 11.132% |
| X-n936-k151 | 132715 | 237625 | 79.049% | 186262 | 40.347% | 220926 | 66.466% | **167584** | **26.274%** |
| X-n957-k87 | 85465 | 130850 | 53.104% | **98198** | **14.898%** | 113882 | 33.250% | 117787 | 37.819% |
| X-n979-k58 | 118976 | 147687 | 24.132% | 138092 | 16.067% | 146347 | 23.005% | **132921** | **11.721%** |
| X-n1001-k43 | 72355 | 100399 | 38.759% | **87660** | **21.153%** | 114448 | 58.176% | 88897 | 22.862% |
| Average Gap | | 29.658% | | 16.769% | | 26.048% | | **16.614%** | |

*Table 15.* Zero-Shot Inference on large-scale CVRP instances from Set-X. Each model is trained on the size n=100, following the setting in (Zhou et al., 2024).

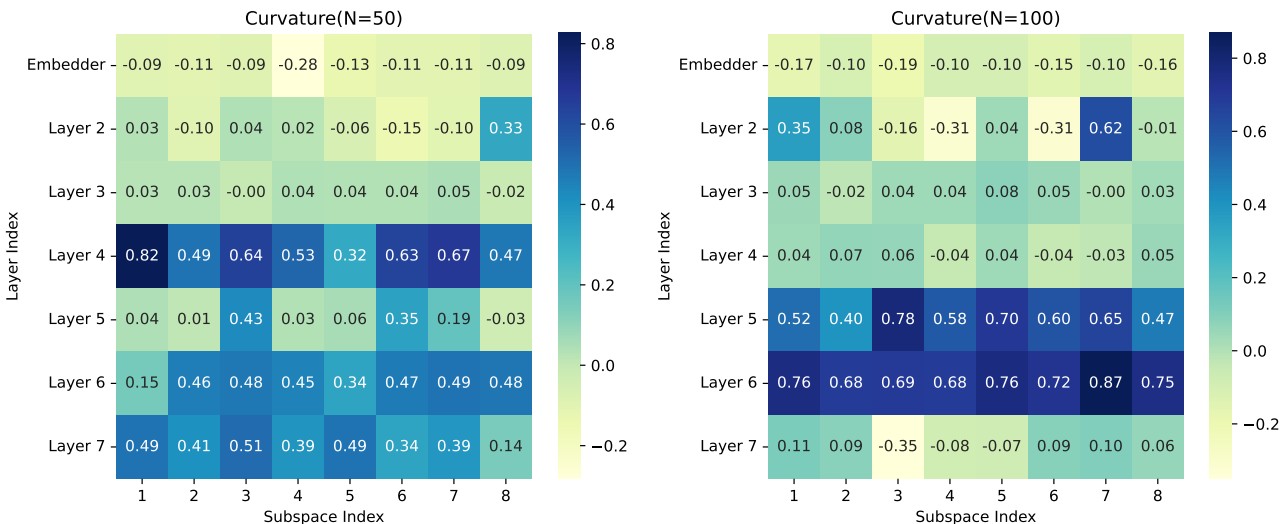

*Figure 6.* Visualization of curvature for each subspace of each layer in the encoder module. Shown model is Mixed-POMO-MTL. The shown colors indicate that subspaces in shallower layers tend to reside in hyperbolic space. As the layer index increases, more subspaces shift closer to spherical geometry. The tendency towards spherical geometry is even more serious when $N = 100$. However, we also observe an unexpected change in last layer where curvatures cluster around zero. Such kind of inconsistency may explain the inferior performances of Mixed-POMO-MTL on some unseen tasks like VRPBTW. Better viewed in color.

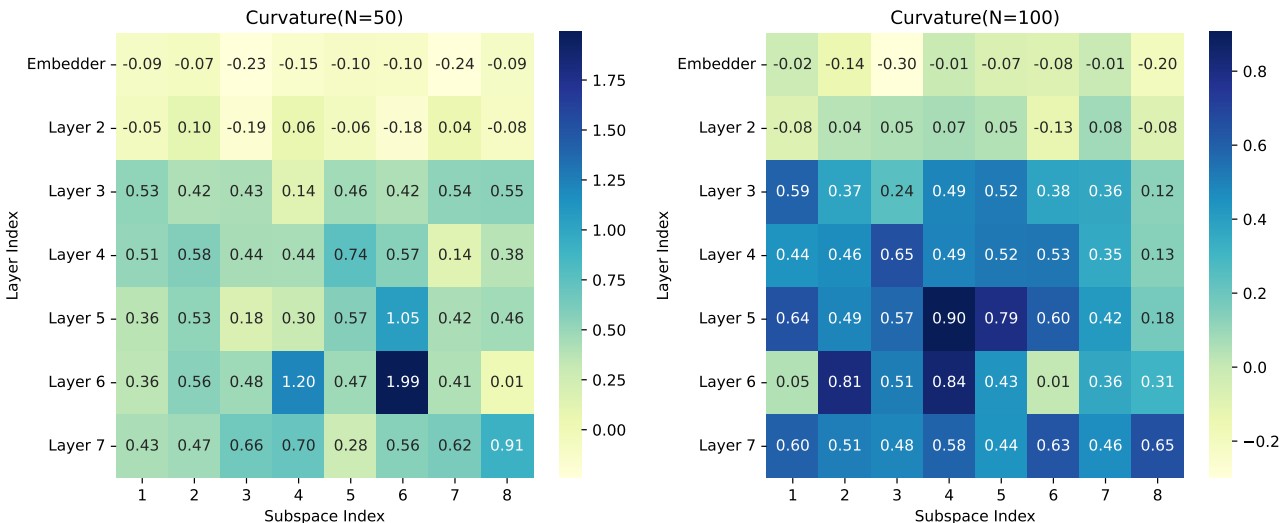

*Figure 7.* Visualization of curvature for each subspace of each layer in the encoder module. Shown model is Mixed-MVMoE-L. The shown colors indicate that subspaces in shallower layers tend to reside in hyperbolic space. As the layer index increases, more subspaces shift closer to spherical geometry. Compared with Mixed-POMO-MTL, it is more consistent as the layer deepens. Better viewed in color.

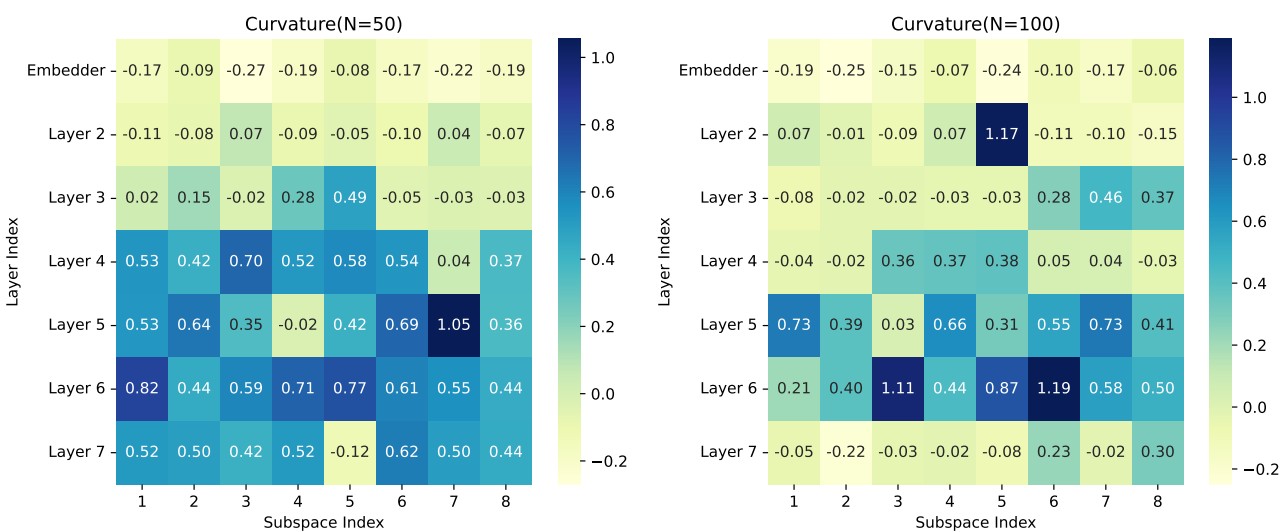

*Figure 8.* Visualization of curvature for each subspace of each layer in the encoder module. Shown model is Mixed-MVMoE (n=100). The shown colors indicate that subspaces in shallower layers tend to reside in hyperbolic space. As the layer index increases, more subspaces shift closer to spherical geometry. As we can observe from the presented color gradients, MVMoE sometimes can learn very positive curvatures even in the shallow layer. We hypothesize this maybe due to the fact that MVMoE doesn't apply approximate routing mechanism so that model itself acquires much stronger abilities to capture high-level information. Better viewed in color.

**Appendix.4. Detailed Experimental Configurations, Results and Visualizations with RouteFinder (Berto et al., 2024)**

**Definitions of Constraints in Utilized Tasks**

In this case, we follow the settings of (Berto et al., 2024) and details are listed as follows:

- **Coordinates:** We focus on uniform distribution setting in which case each node's locations are sampled from $U(0,1)$ in a unit square.

- **Capacity:** We set capacity to 40 and 50 for $n = 50$ and $n = 100$, respectively. Note that one of the hard constraints involved in each task is that nodes with demands greater than dilvery vehicle's current demand are masked.

- **Demand:** We sample the demand of each node from the list $\{1, 2, \ldots, 9\}$. Note that before sending into model, the node demand is normalized by the demand of delivery vehicle.

- **Open Route:** We set it as an indicator vector with all ones. During decoding stage, we need to manually set mask to prevent vehicle from going back to the depot node.

- **Backhauls:** Similar to demand setting, we sample from $\{1, 2, \ldots, 9\}$ as our initial demands. Then, the same as in (Liu et al., 2024), we sample 20% of nodes to be the backhauls nodes.

- **Duration Limit:** We set it to 3, which represents the maximum length of delivery vehicle's route.

- **Time Window:** For the depot, we assign its time window as $[0, 3]$ and service time for depot is 0 by default. However, service time for customer nodes is set to 0.2 and time window for customer nodes are sampled from uniform distribution.

- **Mixed:** In the regular setting, there is a strict preceeding ordering between linehaul and backhaul customers. However, the mixed scenario allows linehaul and backhaul customers to happen in an interleaved manner.

By combining constraints in different ways, we can obtain various kinds of tasks in Table 16. Since some nodes don't have features like time-windows and backhauls, these features will separately appear in the encoder module.

| | Capacity (C) | Open Route (O) | Backhauls (B) | Duration Limit (L) | Time Window (TW) | Mixed (M) |
|---|---|---|---|---|---|---|
| CVRP | ✓ | ✗ | ✗ | ✗ | ✗ | ✗ |
| VRPTW | ✓ | ✗ | ✗ | ✗ | ✓ | ✗ |
| OVRP | ✓ | ✓ | ✗ | ✗ | ✗ | ✗ |
| VRPL | ✓ | ✗ | ✗ | ✓ | ✗ | ✗ |
| VRPB | ✓ | ✗ | ✓ | ✗ | ✗ | ✗ |
| OVRPTW | ✓ | ✓ | ✗ | ✗ | ✓ | ✗ |
| OVRPB | ✓ | ✓ | ✓ | ✗ | ✗ | ✗ |
| OVRPL | ✓ | ✓ | ✗ | ✓ | ✗ | ✗ |
| VRPBL | ✓ | ✗ | ✓ | ✓ | ✗ | ✗ |
| VRPBTW | ✓ | ✗ | ✓ | ✗ | ✓ | ✗ |
| VRPLTW | ✓ | ✗ | ✗ | ✓ | ✓ | ✗ |
| OVRPBL | ✓ | ✓ | ✓ | ✓ | ✗ | ✗ |
| OVRPBTW | ✓ | ✓ | ✓ | ✗ | ✓ | ✗ |
| OVRPLTW | ✓ | ✓ | ✗ | ✓ | ✓ | ✗ |
| VRPBLTW | ✓ | ✗ | ✓ | ✓ | ✓ | ✗ |
| OVRPBLTW | ✓ | ✓ | ✓ | ✓ | ✓ | ✗ |
| VRPMB | ✓ | ✗ | ✓ | ✗ | ✗ | ✓ |
| OVRPMB | ✓ | ✓ | ✓ | ✗ | ✗ | ✓ |
| VRPMBL | ✓ | ✗ | ✓ | ✓ | ✗ | ✓ |
| VRPMBTW | ✓ | ✗ | ✓ | ✗ | ✓ | ✓ |
| OVRPMBL | ✓ | ✓ | ✓ | ✓ | ✗ | ✓ |
| OVRPMBTW | ✓ | ✓ | ✓ | ✗ | ✓ | ✓ |
| VRPMBLTW | ✓ | ✗ | ✓ | ✓ | ✓ | ✓ |
| OVRPMBLTW | ✓ | ✓ | ✓ | ✓ | ✓ | ✓ |

*Table 16.* Detailed descriptions of constraints contained in each problem type. We have 24 VRP tasks in total. Note that the first 16 VRP tasks get involved in training stage and the last 8 tasks are used for few-shot testings.

| Hyper-Parameters | Value |
|---|---|
| Training Epochs | 300 |
| Fine-tuning Epochs | 10 |
| Instances in each Training Epoch | 100,000 |
| Instances in each Fine-tuning Epoch | 10,000 |
| Optimizer | Adam |
| LR Scheduler | MultiStepLR |
| LR Milestones | [270,295] |
| LR Gamma | 0.1 |
| Gradient Clips | 1.0 |
| Training Learning Rate | 3e-4 |
| Fine-tuning Learning Rate | 3e-4 |
| Weight Decay | 1e-6 |
| Training Batch Size | 256 |
| Fine-tuning Batch Szie | 256 |
| Evaluation Batch Size | 128 |
| Problem Scales | {50, 100} |
| Node Distribution | $U(0,1)$ |
| Number of Experts in MoE | 4 |
| Auxiliary Loss Weight in MoE | 0.001 |
| Gating Mechanism in MoE | node-level, input-choice gating |
| Embedding Size | 128 |
| Hidden Feature Size | 512 |
| Number of Encoder Layers | 6 |
| QKV Dimension | 16 |
| Attention Head Number | 8 |
| Logit Clipping | 10 |
| Evaluation Type | argmax |
| Number of Experts in MoE for Routing | 2 |
| Number of Subspaces ($C$) | 8 |
| Initialization value of Curvature ($\kappa$) | 0 |
| Initialization value of $\alpha, \beta$ | {1,1} |

*Table 17.* Detailed experiment settings of hyper-parameters for RouteFinder (Berto et al., 2024) based model. However, other choices for the number of subspaces are also valid as long as the sum of subspaces' dimensions equals 128. Even if 128 is not divisible by number of subspaces, we can still determine dimensions manually or automatically (*e.g.*, neural architecture search (Elsken et al., 2019)).

| Type | Model | Obj | n=50 Gap | Time | Obj | n=100 Gap | Time | Type | Model | Obj | n=50 Gap | Time | Obj | n=100 Gap | Time |
|---|---|---|---|---|---|---|---|---|---|---|---|---|---|---|---|
| CVRP | HGS-PyVRP | 10.372 | 0.000% | 10.4m | 15.628 | 0.000% | 20.8m | VRPTW | HGS-PyVRP | 16.031 | 0.000% | 10.4m | 25.423 | 0.000% | 20.8m |
| | OR-Tools | 10.572 | 1.907% | 10.4m | 16.280 | 4.178% | 20.8m | | OR-Tools | 16.089 | 0.347% | 10.4m | 25.814 | 1.506% | 20.8m |
| | MTPOMO | 10.518 | 1.411% | 2s | 15.934 | 1.988% | 7s | | MTPOMO | 16.410 | 2.364% | 1s | 26.412 | 3.873% | 7s |
| | Mixed-MTPOMO | 10.518 | 1.413% | 2s | 15.951 | 2.095% | 8s | | Mixed-MTPOMO | 16.414 | 2.391% | 2s | 26.388 | 3.780% | 8s |
| | MVMoE | 10.501 | 1.242% | 2s | 15.888 | 1.694% | 9s | | MVMoE | 16.404 | 2.329% | 2s | 26.389 | 3.788% | 9s |
| | Mixed-MVMoE | 10.503 | 1.265% | 3s | 15.887 | 1.690% | 10s | | Mixed-MVMoE | 16.396 | 2.272% | 3s | 26.387 | 3.775% | 10s |
| | RF-MoE | 10.499 | 1.226% | 2s | 15.876 | 1.622% | 9s | | RF-MoE | 16.389 | 2.234% | 2s | 26.322 | 3.519% | 9s |
| | Mixed-RF-MVMoE | 10.500 | 1.230% | 3s | 15.866 | 1.559% | 10s | | Mixed-RF-MVMoE | 16.371 | 2.118% | 3s | 26.307 | 3.457% | 10s |
| | RF-TE | 10.504 | 1.274% | 1s | 15.857 | 1.505% | 7s | | RF-TE | 16.364 | 2.077% | 1s | 26.235 | 3.178% | 7s |
| | Mixed-RF-TE | **10.493** | **1.166%** | 3s | **15.846** | **1.440%** | 9s | | Mixed-RF-TE | **16.320** | **1.798%** | 3s | **26.167** | **2.914%** | 9s |
| OVRP | HGS-PyVRP | 6.507 | 0.000% | 10.4m | 9.725 | 0.000% | 20.8m | VRPL | HGS-PyVRP | 10.587 | 0.000% | 10.4m | 15.766 | 0.000% | 20.8m |
| | OR-Tools | 6.553 | 0.686% | 10.4m | 9.995 | 2.732% | 20.8m | | OR-Tools | 10.570 | 2.343% | 10.4m | 16.466 | 5.302% | 20.8m |
| | MTPOMO | 6.718 | 3.209% | 1s | 10.210 | 4.965% | 6s | | MTPOMO | 10.775 | 1.734% | 1s | 16.149 | 2.434% | 7s |
| | Mixed-MTPOMO | 6.714 | 3.150% | 2s | 10.230 | 5.166% | 8s | | Mixed-MTPOMO | 10.771 | 1.698% | 2s | 16.161 | 2.513% | 8s |
| | MVMoE | 6.702 | 2.965% | 2s | 10.177 | 4.621% | 9s | | MVMoE | 10.751 | 1.505% | 2s | 16.099 | 2.115% | 9s |
| | Mixed-MVMoE | 6.699 | 2.929% | 3s | 10.181 | 4.658% | 9s | | Mixed-MVMoE | 10.752 | 1.523% | 3s | 16.099 | 2.118% | 9s |
| | RF-MoE | 6.697 | 2.886% | 2s | 10.139 | 4.229% | 9s | | RF-MoE | 10.737 | 1.388% | 2s | 16.070 | 1.941% | 9s |
| | Mixed-RF-MVMoE | 6.689 | 2.764% | 3s | 10.137 | 4.216% | 10s | | Mixed-RF-MVMoE | 10.736 | 1.381% | 3s | 16.062 | 1.888% | 9s |
| | RF-TE | 6.684 | 2.687% | 1s | 10.121 | 4.055% | 6s | | RF-TE | 10.749 | 1.502% | 1s | 16.051 | 1.827% | 6s |
| | Mixed-RF-TE | **6.675** | **2.551%** | 2s | **10.111** | **3.946%** | 7s | | Mixed-RF-TE | **10.731** | **1.339%** | 2s | **16.040** | **1.751%** | 7s |
| VRPB | HGS-PyVRP | 9.687 | 0.000% | 10.4m | 14.377 | 0.000% | 20.8m | OVRPTW | HGS-PyVRP | 10.510 | 0.000% | 10.4m | 16.926 | 0.000% | 20.8m |
| | OR-Tools | 9.802 | 1.159% | 10.4m | 14.933 | 3.853% | 20.8m | | OR-Tools | 10.519 | 0.078% | 10.4m | 17.027 | 0.583% | 20.8m |
| | MTPOMO | 10.033 | 3.564% | 1s | 15.082 | 4.922% | 6s | | MTPOMO | 10.668 | 1.479% | 1s | 17.420 | 2.892% | 7s |
| | Mixed-MTPOMO | 10.035 | 3.583% | 2s | 15.100 | 5.045% | 7s | | Mixed-MTPOMO | 10.676 | 1.555% | 2s | 17.419 | 2.889% | 7s |
| | MVMoE | 10.005 | 3.270% | 2s | 15.023 | 4.508% | 9s | | MVMoE | 10.669 | 1.492% | 2s | 17.416 | 2.872% | 10s |
| | Mixed-MVMoE | 10.002 | 3.242% | 2s | 15.027 | 4.537% | 10s | | Mixed-MVMoE | 10.665 | 1.459% | 2s | 17.393 | 2.738% | 10s |
| | RF-MoE | 9.980 | 3.015% | 2s | 14.973 | 4.164% | 8s | | RF-MoE | 10.674 | 1.539% | 2s | 17.387 | 2.697% | 10s |
| | Mixed-RF-MVMoE | 9.980 | 3.012% | 2s | 14.962 | 4.085% | 9s | | Mixed-RF-MVMoE | 10.660 | 1.403% | 2s | 17.369 | 2.592% | 11s |
| | RF-TE | 9.977 | 2.989% | 1s | 14.942 | 3.952% | 6s | | RF-TE | 10.652 | 1.326% | 1s | 17.327 | 2.346% | 7s |
| | Mixed-RF-TE | **9.963** | **2.832%** | 2s | **14.929** | **3.863%** | 7s | | Mixed-RF-TE | **10.635** | **1.166%** | 2s | **17.285** | **2.100%** | 7s |
| VRPBL | HGS-PyVRP | 10.186 | 0.000% | 10.4m | 14.779 | 0.000% | 20.8m | VRPBLTW | HGS-PyVRP | 15.510 | 0.000% | 10.4m | 16.926 | 0.000% | 20.8m |
| | OR-Tools | 10.331 | 1.390% | 10.4m | 15.426 | 4.338% | 20.8m | | OR-Tools | 18.422 | 0.332% | 10.4m | 29.830 | 2.770% | 20.8m |
| | MTPOMO | 10.672 | 4.697% | 1s | 15.712 | 6.251% | 7s | | MTPOMO | 18.990 | 2.128% | 1s | 30.898 | 3.624% | 7s |
| | Mixed-MTPOMO | 10.666 | 4.644% | 2s | 15.728 | 6.359% | 8s | | Mixed-MTPOMO | 19.015 | 2.258% | 2s | 30.897 | 3.616% | 8s |
| | MVMoE | 10.637 | 4.354% | 2s | 15.640 | 5.758% | 9s | | MVMoE | 18.985 | 2.100% | 2s | 30.892 | 3.608% | 10s |
| | Mixed-MVMoE | 10.640 | 4.394% | 2s | 15.647 | 5.816% | 10s | | Mixed-MVMoE | 18.977 | 2.060% | 2s | 30.883 | 3.569% | 11s |
| | RF-MoE | 10.575 | 3.765% | 2s | 15.541 | 5.121% | 9s | | RF-MoE | 18.957 | 1.960% | 2s | 30.808 | 3.323% | 10s |
| | Mixed-RF-MVMoE | **10.568** | **3.702%** | 2s | 15.537 | 5.089% | 10s | | Mixed-RF-MVMoE | 18.939 | 1.873% | 2s | 30.773 | 3.202% | 11s |
| | RF-TE | 10.578 | 3.803% | 1s | 15.528 | 5.039% | 6s | | RF-TE | 18.941 | 1.877% | 1s | 30.688 | 2.923% | 7s |
| | Mixed-RF-TE | **10.553** | **3.555%** | 2s | **15.499** | **4.843%** | 7s | | Mixed-RF-TE | **18.894** | **1.621%** | 2s | **30.642** | **2.768%** | 8s |
| VRPBTW | HGS-PyVRP | 18.292 | 0.000% | 10.4m | 29.467 | 0.000% | 20.8m | VRPLTW | HGS-PyVRP | 16.356 | 0.000% | 10.4m | 25.757 | 0.000% | 20.8m |
| | OR-Tools | 18.366 | 0.383% | 10.4m | 29.945 | 1.597% | 20.8m | | OR-Tools | 16.441 | 0.499% | 10.4m | 26.259 | 1.899% | 20.8m |
| | MTPOMO | 18.639 | 1.878% | 1s | 30.437 | 3.285% | 7s | | MTPOMO | 16.824 | 2.823% | 1s | 26.891 | 4.368% | 7s |
| | Mixed-MTPOMO | 18.659 | 1.985% | 2s | 30.428 | 3.253% | 8s | | Mixed-MTPOMO | 16.816 | 2.779% | 2s | 26.882 | 4.330% | 8s |
| | MVMoE | 18.640 | 1.883% | 2s | 30.436 | 3.281% | 9s | | MVMoE | 16.811 | 2.750% | 2s | 26.868 | 4.277% | 9s |
| | Mixed-MVMoE | 18.630 | 1.830% | 2s | 30.422 | 3.232% | 10s | | Mixed-MVMoE | 16.804 | 2.703% | 2s | 26.851 | 4.211% | 10s |
| | RF-MoE | 18.616 | 1.757% | 2s | 30.341 | 2.954% | 9s | | RF-MoE | 16.777 | 2.550% | 2s | 26.774 | 3.912% | 9s |
| | Mixed-RF-MVMoE | 18.607 | 1.706% | 2s | 30.306 | 2.839% | 10s | | Mixed-RF-MVMoE | 16.762 | 2.453% | 2s | 26.746 | 3.802% | 10s |
| | RF-TE | 18.600 | 1.676% | 1s | 30.241 | 2.619% | 7s | | RF-TE | 16.762 | 2.454% | 1s | 26.689 | 3.579% | 7s |
| | Mixed-RF-TE | **18.555** | **1.417%** | 8s | **30.172** | **2.385%** | 8s | | Mixed-RF-TE | **16.706** | **2.121%** | 2s | **26.637** | **3.377%** | 8s |
| OVRPB | HGS-PyVRP | 6.898 | 0.000% | 10.4m | 10.335 | 0.000% | 20.8m | OVRPBL | HGS-PyVRP | 6.899 | 0.000% | 10.4m | 10.335 | 0.000% | 20.8m |
| | OR-Tools | 6.928 | 0.412% | 10.4m | 10.577 | 2.315% | 20.8m | | OR-Tools | 6.927 | 0.386% | 10.4m | 10.582 | 2.363% | 20.8m |
| | MTPOMO | 7.108 | 3.005% | 1s | 10.878 | 5.224% | 7s | | MTPOMO | 7.112 | 3.055% | 1s | 10.884 | 5.276% | 6s |
| | Mixed-MTPOMO | 7.099 | 2.889% | 2s | 10.892 | 5.354% | 8s | | Mixed-MTPOMO | 7.108 | 3.002% | 2s | 10.899 | 5.419% | 8s |
| | MVMoE | 7.089 | 2.741% | 2s | 10.840 | 4.861% | 9s | | MVMoE | 7.098 | 2.846% | 2s | 10.847 | 4.928% | 9s |
| | Mixed-MVMoE | 7.088 | 2.729% | 2s | 10.835 | 4.809% | 10s | | Mixed-MVMoE | 7.090 | 2.739% | 2s | 10.842 | 4.878% | 10s |
| | RF-MoE | 7.080 | 2.513% | 2s | 10.805 | 4.522% | 9s | | RF-MoE | 7.083 | 2.635% | 2s | 10.806 | 4.534% | 9s |
| | Mixed-RF-MVMoE | 7.075 | 2.509% | 2s | 10.791 | 4.388% | 10s | | Mixed-RF-MVMoE | 7.076 | 2.539% | 2s | 10.796 | 4.428% | 10s |
| | RF-TE | 7.071 | 2.479% | 1s | 10.772 | 4.208% | 7s | | RF-TE | 7.074 | 2.508% | 1s | 10.778 | 4.262% | 7s |
| | Mixed-RF-TE | **7.053** | **2.216%** | 2s | **10.745** | **3.939%** | 8s | | Mixed-RF-TE | **7.054** | **2.215%** | 2s | **10.749** | **3.979%** | 8s |
| OVRPBLTW | HGS-PyVRP | 11.668 | 0.000% | 10.4m | 19.156 | 0.000% | 20.8m | OVRPBTW | HGS-PyVRP | 11.669 | 0.000% | 10.4m | 19.15 | 0.000% | 20.8m |
| | OR-Tools | 11.681 | 0.106% | 10.4m | 19.305 | 0.767% | 20.8m | | OR-Tools | 11.682 | 0.109% | 10.4m | 19.303 | 0.757% | 20.8m |
| | MTPOMO | 11.817 | 1.260% | 1s | 19.637 | 2.496% | 7s | | MTPOMO | 11.814 | 1.229% | 1s | 19.635 | 2.485% | 7s |
| | Mixed-MTPOMO | 11.823 | 1.312% | 2s | 19.634 | 2.476% | 8s | | Mixed-MTPOMO | 11.824 | 1.315% | 2s | 19.631 | 2.465% | 8s |
| | MVMoE | 11.822 | 1.301% | 2s | 19.641 | 2.518% | 10s | | MVMoE | 11.819 | 1.271% | 2s | 19.638 | 2.503% | 10s |
| | Mixed-MVMoE | 11.814 | 1.228% | 2s | 19.621 | 2.412% | 10s | | Mixed-MVMoE | 11.813 | 1.216% | 2s | 19.624 | 2.424% | 11s |
| | RF-MoE | 11.824 | 1.312% | 2s | 19.607 | 2.334% | 10s | | RF-MoE | 11.823 | 1.304% | 2s | 19.606 | 2.328% | 10s |
| | Mixed-RF-MVMoE | 11.813 | 1.225% | 2s | 19.584 | 2.212% | 11s | | Mixed-RF-MVMoE | 11.813 | 1.218% | 2s | 19.583 | 2.209% | 11s |
| | RF-TE | 11.805 | 1.150% | 1s | 19.551 | 2.048% | 7s | | RF-TE | 11.805 | 1.151% | 1s | 19.550 | 2.042% | 7s |
| | Mixed-RF-TE | **11.780** | **0.949%** | 2s | **19.503** | **1.793%** | 8s | | Mixed-RF-TE | **11.780** | **0.946%** | 2s | **19.501** | **1.783%** | 8s |

| Type | Model | Obj | n=50 Gap | Time | Obj | n=100 Gap | Time | Type | Model | Obj | n=50 Gap | Time | Obj | n=100 Gap | Time |
|---|---|---|---|---|---|---|---|---|---|---|---|---|---|---|---|
| OVRPL | HGS-PyVRP | 6.507 | 0.000% | 10.4m | 9.724 | 0.000% | 20.8m | OVRPLTW | HGS-PyVRP | 10.510 | 0.000% | 10.4m | 16.926 | 0.000% | 20.8m |
| | OR-Tools | 6.552 | 0.668% | 10.4m | 10.001 | 2.791% | 20.8m | | OR-Tools | 10.497 | 0.114% | 10.4m | 17.023 | 0.728% | 20.8m |
| | MTPOMO | 6.719 | 3.227% | 1s | 10.214 | 5.002% | 6s | | MTPOMO | 10.670 | 1.500% | 1s | 17.420 | 2.889% | 7s |
| | Mixed-MTPOMO | 6.715 | 3.159% | 2s | 10.234 | 5.214% | 7s | | Mixed-MTPOMO | 10.678 | 1.571% | 2s | 17.418 | 2.882% | 8s |
| | MVMoE | 6.707 | 3.030% | 2s | 10.184 | 4.696% | 9s | | MVMoE | 10.671 | 1.511% | 2s | 17.419 | 2.885% | 10s |
| | Mixed-MVMoE | 6.700 | 2.949% | 2s | 10.183 | 4.683% | 10s | | Mixed-MVMoE | 10.662 | 1.429% | 2s | 17.397 | 2.759% | 11s |
| | RF-MoE | 6.696 | 2.864% | 2s | 10.140 | 4.249% | 9s | | RF-MoE | 10.673 | 1.532% | 2s | 17.386 | 2.693% | 10s |
| | Mixed-RF-MVMoE | 6.689 | 2.762% | 2s | 10.136 | 4.202% | 10s | | Mixed-RF-MVMoE | 10.661 | 1.413% | 2s | 17.369 | 2.591% | 11s |
| | RF-TE | 6.686 | 2.721% | 1s | 10.120 | 4.052% | 6s | | RF-TE | 10.653 | 1.341% | 1s | 17.327 | 2.347% | 7s |
| | Mixed-RF-TE | **6.675** | **2.545%** | 2s | **10.110** | **3.937%** | 7s | | Mixed-RF-TE | **10.636** | **1.176%** | 2s | **17.287** | **2.108%** | 8s |

Table 18: Each model's performances on 16 seen tasks following the setting of (Berto et al., 2024). Each task is assigned with 1,000 instances for testing. The best performances are annotated with bold and domains improved by our module are highlighted with underlines.

| VRPLib | RF-POMO-MTL Gap | RF-MVMoE Gap | RF-TE Gap | Mixed-RF-TE Gap |
|---|---|---|---|---|
| A | 2.529% | 2.833% | 2.825% | 2.454% |
| B | 2.752% | 3.171% | 2.583% | 2.665% |
| E | 5.069% | 2.348% | 2.929% | 3.369% |
| F | 12.772% | 14.858% | 12.951% | 11.479% |
| M | 5.907% | 7.010% | 5.078% | 5.102% |
| P | 4.678% | 3.389% | 4.573% | 4.254% |
| X | 9.143% | 10.259% | 8.435% | 9.458% |
| Average Gap | 6.121% | 6.267% | 5.627% | **5.579%** |

*Table 19.* Zero-Shot Inference on CVRP benchmark instances from Set-X (Uchoa et al., 2017).

| Method | VRPMB Cost | VRPMB Gap | OVRPMB Cost | OVRPMB Gap | VRPMBL Cost | VRPMBL Gap | VRPMBTW Cost | VRPMBTW Gap | OVRPMBL Cost | OVRPMBL Gap | OVRPMBTW Cost | OVRPMBTW Gap | VRPMBLTW Cost | VRPMBLTW Gap | OVRPMBLTW Cost | OVRPMBLTW Gap |
|---|---|---|---|---|---|---|---|---|---|---|---|---|---|---|---|---|
| HGS-PyVRP | 13.54 | 0.00% | 9.01 | 0.00% | 13.78 | 0.00% | 25.51 | 0.00% | 9.01 | 0.00% | 16.97 | 0.00% | 25.85 | 0.00% | 16.97 | 0.00% |
| OR-Tools | 14.93 | 10.27% | 10.59 | 17.54% | 15.42 | 11.90% | 29.97 | 17.48% | 10.59 | 17.54% | 19.31 | 13.78% | 30.44 | 17.76% | 19.31 | 13.78% |
| Zero-shot | 14.88 | 10.13% | 10.72 | 19.02% | 15.18 | 10.32% | 28.29 | 10.89% | 10.72 | 19.01% | 18.45 | 8.68% | 28.65 | 10.82% | 18.45 | 8.69% |
| Mixed-Zero-Shot | 16.04 | 19.12% | 12.62 | 31.20% | 16.61 | 20.95% | 28.73 | 12.71% | 11.72 | 30.40% | 18.75 | 10.55% | 29.40 | 13.78% | 18.78 | 10.72% |
| Train (scratch) | 15.12 | 12.13% | 10.40 | 15.35% | 16.32 | 18.27% | 28.15 | 10.71% | 10.18 | 16.08% | 18.36 | 11.19% | 28.69 | 10.95% | 18.86 | 11.19% |
| Mixed (scratch) | 14.60 | 7.96% | 9.61 | 6.65% | 14.83 | 7.61% | 26.56 | 4.18% | 9.62 | 6.76% | 17.54 | 3.37% | 26.98 | 4.47% | 17.54 | 3.39% |
| EAL (step 0) | 14.88 | 10.13% | 10.72 | 19.02% | 15.18 | 10.32% | 28.29 | 10.89% | 10.72 | 19.01% | 18.45 | 8.68% | 28.65 | 10.82% | 18.45 | 8.69% |
| Mixed-EAL (step 0) | 16.04 | 19.12% | 12.62 | 31.20% | 16.61 | 20.95% | 28.73 | 12.71% | 11.72 | 30.40% | 18.75 | 10.55% | 29.40 | 13.78% | 18.78 | 10.72% |
| EAL | 14.59 | 7.89% | 9.66 | 7.19% | 14.78 | 7.39% | 26.69 | 4.61% | 9.65 | 7.13% | 17.59 | 3.65% | 27.13 | 4.90% | 17.59 | 3.65% |
| Mixed-EAL | **14.03** | **3.68%** | **9.37** | **4.02%** | **14.31** | **3.89%** | **26.46** | **3.69%** | **9.37** | **4.06%** | **17.46** | **2.86%** | **26.89** | **3.99%** | **17.45** | **2.84%** |

*Table 20.* Performance comparisons under few-shot scenario on 8 unseen tasks. EAL here denotes the Efficient Adapter Layer proposed in (Berto et al., 2024). It pads zeros on original weight matrix, which can infuse unseen features into model. Following (Berto et al., 2024), each model is trained on the size $N = 100$ and each task is assigned with 1,000 test instances for validations. During few-shot learning, model is trained with 10 epochs and each epoch contains 10,000 instances. The best performances are annotated with bold and domains improved by our module are highlighted with underlines.

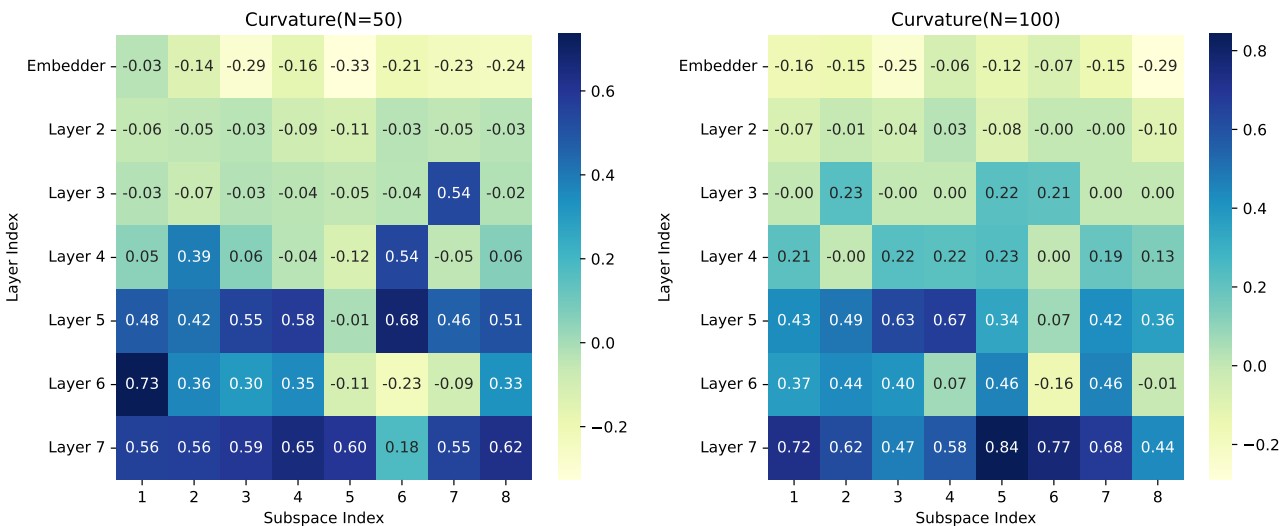

*Figure 9.* Visualization of curvature for each subspace of each layer in the encoder module. Shown model is Mixed-RF-MVMoE. The shown colors indicate that subspaces in shallower layers tend to reside in hyperbolic space. As the layer index increases, more subspaces shift closer to spherical geometry. Similar to the results presented in Figure 8, the MVMoE based model keeps the consistency in the evolution of curvatures across layers. Better viewed in color.

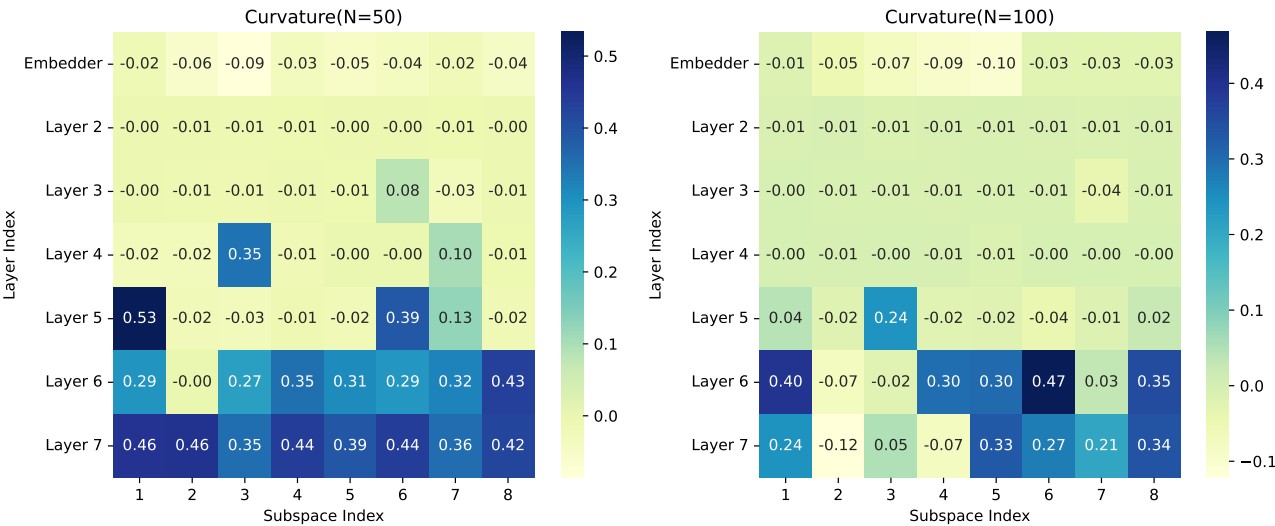

*Figure 10.* Visualization of curvature for each subspace of each layer in the encoder module. Shown model is Mixed-RF-TE. The shown colors indicate that subspaces in shallower layers tend to reside in hyperbolic space. As the layer index increases, more subspaces shift closer to spherical geometry. Compared to the MVMoE based models in Figure 8, Mixed-RF-TE doesn't acquire very positive or negative curvatures, they mostly cluster around the zero point. Better viewed in color.

