# OpenReview forum: "A Mixed-Curvature based Pre-training Paradigm for Multi-Task Vehicle Routing Solver"
_ICML.cc/2025/Conference — ICML 2025 poster_

### Official Review · Reviewer_gmJz · 2025-03-09

**Overall Recommendation:** 3

**Summary:**

In the context of neural solvers for vehicle routing problems (VRP), the paper introduces a methodology to learn embeddings in non-Euclidean space, where the embedding space is divided into several subspaces with adaptively learned curvatures.

**Claims And Evidence:**

The authors claim that their proposition can be integrated in various existing solvers and improve their performance. This claim is empirically demonstrated with three backbone methods (i.e., POMO-MTL, MVMoE, RouteFinder) on various problems (e.g., random and real instances) in different settings (e.g., in-distribution, zero-shot, few-shot test).

I would have appreciated an ablation study (e.g., for mix-up technique) and/or sensitivity analysis (e.g., number of subspaces) for the proposed method to have a stronger confidence in the results.

**Essential References Not Discussed:**

I'm not aware of a missing reference.

**Experimental Designs Or Analyses:**

The empirical validation uses standard evaluation criteria (e.g., gap) and is conducted according to different previously-proposed protocols. The analysis followed by the authors is standard when evaluating neural solvers. It would have been nice to report computational times and the sizes of the models.

**Methods And Evaluation Criteria:**

The proposed method defines a more expressive neural architecture where embeddings are divided in several subspaces with different curvatures. The architecture seems natural, although the need of the mix-up technique (equation 13) seems a bit strange to me, since it amounts to do a linear combination of two embeddings in two different spaces. The use of this technique is only justified empirically (i.e., it leads to better performance). The explanation in the first sentence of page 5, col. 2 is not very convincing to me. Why is the transition non-smooth?

**Other Comments Or Suggestions:**

The paper should be proofread, e.g.,
- the meaning of acronyms should ideally be recalled.
- there are many typos in page 3.
- in page 4, the cardinal of V is actually n+1.
- the bold values seem to be off in Table 6 in the appendix.
- It would be nice if there were some comments on the figures showing the curvatures in the appendix.

**Other Strengths And Weaknesses:**

The idea of learning embeddings in non-Euclidean spaces with different curvatures for solving VRP is novel. It seems that it may introduce more parameters and the computation time increases. This discussion is missing in the current paper.

**Questions For Authors:**

1. Why is the transition between different curvature spaces non-smooth?
2. How did you choose the hyperparameters specific to your proposed method?
3. How does the number of parameters changes when using your proposed architecture compared to a backbone architecture?
4. What is the additional computational cost required by your proposed method?

**Relation To Broader Scientific Literature:**

The paper summarizes recent research in solving VRP and in deep learning in non-Euclidean space. The work in this paper is at the intersection of those areas. As far as I know, the relevant work seems to be discussed.

**Theoretical Claims:**

There is no theoretical claim in this paper.

---

> ### Author Rebuttal · Authors · 2025-04-01
>
> We appreciate the reviewer gmJz's insightful questions and constructive feedbacks.
> #### **[Acronyms]** Thank you for the suggestion. We will ensure all acronyms are properly defined upon first use and provide a reference table in the final version. For example, "VRPs" stands for Vehicle Routing Problems, "LKH" for Lin-Kernighan-Helsgaun and "HGS" for Hybrid Genetic Search, etc.
> #### **[Typos]** We appreciate the attention to typos. We identified and will correct errors such as "manioflds" (line 123), "hypersphereical" (line 124), "$n$ nodes" (line 185). The cardinal of $V$ on page 4 is indeed $n+1$ and we will correct it. Besides, in Table 6 (Appendix.3), we notice that for the row indexed by "R102", two columns are mis-highlighted at same time. We have checked all tables and will make sure all bold values are correctly used.
> #### **[Comments on curvature figure]** In Figure 4 (Appendix.1), we show node curvatures on 10 OOD tasks. It shows that almost every node in the dataset has either negative or positive curvature and the average curvature suggests that each task  contains geometry patterns. For Figure 5-10 (Appendix.3), we provided corresponding comments in "curvature analysis" part (Appendix.1), where shallow layers and deeper layers present different curvatures. We will move this analysis directly into these figure captions in the final version.
> #### **[Why is transition non-smooth]** The transition between different curvature spaces is non-smooth primarily due to the incompatibility of geometric structures among different subspaces. In specifics, hyperbolic and spherical spaces have fundamentally different distance metrics. In details, the geodesic distance in hyperbolic space grows exponentially when $|\kappa|$ ($\kappa$ here denotes negative curvature of hyperbolic space) increases. While on the other hand, the distance in spherical space is bounded. This discrepancy may introduce unexpected disturbance in feature transformation. In Figure 5-10 (Appendix.3), we visualize the curvature information of different subspaces layer by layer. From the results in these figures, the shallow layers tend to process features in hyperbolic spaces while deeper layers choose to process features via spherical spaces. The transitions between hyperbolic space and spherical space usualy happen around layer 2 and layer 3. The inferior features caused by non-smooth transition will severly affect deeper layers' learning ability. To mitigate this, we apply techniques similar to Mix-Up[1] where previous layer's features was infused into deeper layers. This can guide deeper layers gradually move into new curvature spaces.
>
> #### **[Settings of hyperparameters]** Table 5 (Appendix.2) lists our detailed settings of hyperparameters and we strictly follow prior works[2,3]: 5,000 training epochs, batch size of 128, and 20,000 instances per epoch from 6 VRPs. Fine-tuning uses 10,000 instances per epoch for 10 epochs with the same batch size. We employ Adam optimizer with learning rate 1e-4, decayed by 10 after 4,500 epochs. Feature space (128 dimensions) is split into 8 subspaces of 16 dimensions, initialized at 0. Number of attention heads and encoder layers are 8 and 6, respectively.
>
> #### **[Number of parameters in our method]** For embedder, we insert two mixed-curvature  modules for depot and customer node, respectively. For each layer of encoder, we insert one mixed-curvature module for processing feautres from previous layer. The comparisons between baselines and mixed-curvature versions are listed as follows:
> | **Models** | **Number of parameters** |
> |---------------------------------------------|-------------------------------------|
> | POMO-MTL | 1,254,656 |
> | Mixed-POMO-MTL | 1,386,810 |
> | MVMOE-Light | 3,698,944  |
> | Mixed-MVMOE-Light | 3,831,116 |
> | MVMOE | 3,682,176 |
> | Mixed-MVMOE | 3,814,348|
> #### From table, compared with original backbones, the number of parameters is inreased by 3.57\%~10.56\%. We will emphasize these changes in the final version.
>
> #### **[Computational cost]** To compare the computational cost with baseline models like POMO-MTL and MVMOE, we list the running time **[here](https://anonymous.4open.science/r/15699-8B57/README.md)**. From results in the table, running time of our model increases a little bit compared to backbones. Taking (n=100) as example, Mixed-POMO-MTL, Mixed-MVMOE-L and Mixed-MVMOE introduce 10.72\%, 7.68\%, 7.56\% costs to backbones, respectively. These show that mixed-curvature modules bring moderate costs, which would not cause serious inference burden.
> #### **[Ablation]** We conduct ablation studies on MVMOE and results are shown **[here](https://anonymous.4open.science/r/15699-8B57/README.md)**. We can see that Mix-Up can improve performances.
>
> [1]Pomo: Policy optimization with multiple optima for reinforcement learning. NIPS, 2020.
>
> [2]Mvmoe: Multi-task vehicle routing solver with mixture-of-experts. ICML, 2024.
>
> [3]mixup: Beyond empirical risk minimization. ICLR, 2018.

---

### Official Review · Reviewer_qtXz · 2025-03-12

**Overall Recommendation:** 2

**Summary:**

This article considers modeling the VRP tasks in multi-task NCO in Riemannian space instead of the original Euclidean one. This article modifies the embedding layer based on this idea, and the experiment demonstrated the effectiveness of the proposed method.

## update after rebuttal

I think the topic of this article is interesting and super novel, and I have no problem with the possible admission decision. However, since the experiment added by the author does not fully demonstrate the significant effect of non-Euclidean transformation, I am not motivated enough to improve my score.

**Claims And Evidence:**

To my understanding, discussing the relation curvature shown in Figure 1 with the Euclidean space should have at least three dimensions of $ xin X$. Unfortunately, based on the current version of this article, I am not sure whether the $X$ space represents coordinates (2-dimensional) or coordinates with node features (such as time window or capacity). I suspect your intention should be the latter. I suggest you pay attention to this point and add clear illustrations and definitions.

**Essential References Not Discussed:**

N/A

**Experimental Designs Or Analyses:**

What do you mean about ``we use an instance augmentation method for solution decoding.`` 8-augmentation or something else. Please state clearly.

**Methods And Evaluation Criteria:**

I am not familiar with the basic definitions of Riemann manifold and tank space, and I am curious about the specific formulas for the $Exp$ operator and the $Log$ operator in this work.

**Other Comments Or Suggestions:**

Figure 2 is not a vector diagram, please replace it with a vector diagram.

**Other Strengths And Weaknesses:**

**Strength:**
1. The general representation of multi-task NCO features can be improved, and this paper notices such drawback.


**Weakness:**

1. Based on my background, I have limited knowledge of Riemannian space so I am confused about the analysis presented in this article. Based on the current version, I cannot intuitively understand the necessity of mapping features to Riemannian space. Figure 1 attempts to demonstrate the necessity, but this is only a statistical value. In my opinion, it can only indicate that these points are relatively clustered or scattered, and have no strong relationship with whether they belong to the Riemann space or not. Furthermore, the author did not provide any intuitive explanation. After using the method proposed in this article, why does the embedding method proposed in this article achieve better results?

I am looking forward to the author's complete explanation of these motivational doubts.

**Questions For Authors:**

1. Can you please provide me with an intuitive reason why the embedding method proposed in this article can achieve better results?

2. What is the detailed rule for subspace partition when D cannot be divided by C?

**Relation To Broader Scientific Literature:**

I think it is valuable to discuss the distribution of the entire node feature (including constraints and coordinates) for multi-task NCO, and this article first discusses this part.

**Theoretical Claims:**

This paper does not contain theoretical claims.

---

> ### Author Rebuttal · Authors · 2025-04-01
>
> We would like to express our sincere gratitude to the reviewer qtXz for detailed and valuable suggestions and comments.
> #### **[Dimension of $X$]** Since graph is discrete, Ollivier-Ricci metric is used for measuring curvature based on edges weights, which doesn't require dimensions of inputs. This is different from curvature on 3-dimension smooth manifolds. Our $X$ only consists of two dimensions (coordinates) for calculating curvatures. Since some nodes don't have features like time-windows and backhauls, these features will separately appear in encoder.  We will make this clear in the final version.
> #### **[Instance augmentation method]** We follow POMO-MTL[1] and MVMOE[2], applying greedy rollout with 8× instance augmentation for fair comparisons, where best solutions for each instance are obtained by solving multiple (8×) equivalent instances. Those equivalent instances are acquired by rotating or clipping the original instances[7]. We will clarify this in the final version.
> #### **[Vector diagram]** We have created a new vector diagram (**[here](https://anonymous.4open.science/r/VD-5D0E/README.md)**) that will replace the original Figure 2 in the final version.
> #### **[Clarification for motivations]** Following [1,2], settings of VRPs are based on Euclidean space. However, the underlining data structures are graphs and they produce certain kinds of complex structures (e.g., node clustering, scattering) which can be approximated by tree-like or cycle-like patterns[4,5,6]. As pointed out in[4,6], to capture these nuances and variations from datas, Riemannian manifolds provide us with a more flexible and effective framework. Though graph is discrete, which is usually different from Riemannian manifolds that are smooth and continuous, we can use some discrete analogs (e.g., the Ollivier-Ricci curvature used for visualizations in Figure 1) to get some senses.
> #### Recall definitions in Equation 15,16 (Appendix.1), the node curvature is calculated based on edge weights of neighbour nodes. So, given a graph $G$ and two nodes $u$ and $v$, the positive Ollivier-Ricci curvature $\kappa(u,v)$ indicates that $u$ and $v$ share many common points in their neighbours. In other words, if random walks $A$ and $B$ start from $u$ and $v$ respectively, then $A$ and $B$ are more likely to meet with each other in a few steps.  In this case, paths on $G$ tend to collapse together. Similarly, the negative curvature indicates that paths on $G$ tend to diverge and spread out. Such kinds of behaviours are quite similar to those of geodesics on Riemannian manifolds where initially parallel geodesics will converge and diverge on Spherical and Hyperbolic Spaces, respectively. From these, Ollivier-Ricci curvature can be treated as a discrete analogy of curvature on continuous domain and its outputs strongly reflect the existence of non-uniform structures in datas, which motivates us to leverage Riemannian manifold spaces for creating a feature space that preserves the underlying node distances and relative positions more faithfully.
> #### From Table 3 (Appendix.3), the calculated distortion rates (defined in Equation 9) of different models show that Riemannian manifolds have done a better job of preserving original distances among nodes in feature space compared with other baselines, which is more desirable since neural solvers for VRPs heavily rely on the quality of produced hidden features to select next node.
> #### To summarize, graphs used in VRPs often contain intricate relations that can't directly be viewed as flat structures. Though graph is discrete, we can utilize neural networks to firstly embed graph into continuous embeddings and apply curved manifolds to fit the underlining geometric structures, which can let model enjoy much stronger abilities to detect and capture fine-grained information from the inputs.
> #### **[How to set $D$ and $C$ in indivisible case]** In our case, $D=128$ and $C=8$ so each subspace has 16 dimensions. When $D$ is not divisible by $C$, we need to manually design the dimension of each subspace. For instance, when $D=127$ and $C=8$, the first 7 subspaces can have 16 dimensions and last one has 15 dimensions (other choices are also allowed as long as the summation of subspaces' dimensions is equal to $D$). We follow POMO-MTL[1] and MVMOE[2] where $D$ is always divisible by $C$. But this can be explored in future (e.g., using neural architecture search[3] to decide dimensions of subspaces).
>
> [1]Multi-task learning for routing problem with cross-problem zero-shot generalization. KDD, 2024.
>
> [2]Mvmoe: Multi-task vehicle routing solver with mixture-of-experts. ICML, 2024.
>
> [3]Neural architecture search: A survey. JMLR, 2019.
>
> [4]Learning mixed-curvature representations in product spaces. ICLR, 2018.
>
> [5]Hyperbolic graph neural networks. NIPS, 2019.
>
> [6]Constant curvature graph convolutional networks. ICML, 2020.
>
> [7]Pomo: Policy optimization with multiple optima for reinforcement learning. NIPS, 2020.

---

> > ### Comment · Reviewer_qtXz · 2025-04-05
> >
> > Thank you for your reply. Based on my current understanding, the motivation of this article is to use knowledge from non Euclidean space to learn better representations of certain types of complex structures in two-dimensional coordinates. I acknowledge the novelty of this motivation.
> >
> > However, I still have two main concerns that have not been resolved:
> > 1. Can you use some experiments to prove that the complex structure is better represented by the Mixed-MVMoE? It seems that images 5-9 attempt to illustrate this point, but I cannot understand the author's conclusion ``By contrast, as the layer index increases, more subspaces shift closer to spherical geometry``.
> > 2. In the experiment, does the Mixed-MVMoE proposed in this article increase the number of parameters compared to MVMOE? If so, how much has it increased, and is this increase in parameter related to the improvement in effectiveness?
> >
> >
> > > Second Round:
> >
> > Thank you for your reply. My concern 1 has been resolved. However, I think the current experimental results are not sufficient to completely eliminate the doubt that the effectiveness of the proposed method is largely due to the increase in parameters. I will keep my score.

---

> > > ### Author Response · Authors · 2025-04-08
> > >
> > > #### **[Second Round]** Thank you for your second-round feedback. Could you please clarify concerns about the lack of sufficiency? Since deadline hasn't passed yet, we are happy to provide more details if needed.
> > > ---
> > > #### **[Complex structures]** The experiment that demonstrates the ability of our model to keep complex structures is in Table 3 (Appendix.1). In details, we use the distortion rate defined in Eqs. 8 and 9 to measure the difference between distance of two nodes in feature space and that in the original input graphs. For the sake of convenience, we list the definitions of distortion rate and results in Table 3 here:
> > > $$
> > > D_{\text{avg}} = \frac{1}{N} \sum_{a \ne b} \left| \frac{d_{U_1}(f(a), f(b))}{d_{U_2}(a, b)} - 1 \right|,
> > > $$
> > > #### where $U_1,U_2$ denote feature space and input graph metric space, respectively. For distoratoin rate, the lower the better.
> > > | Model            | Distortion |
> > > |------------------|------------|
> > > | POMO-MTL         | 2477.725   |
> > > | MVMoE-L          | 2923.151   |
> > > | MVMoE            | 2083.015   |
> > > | Mixed-POMO-MTL   | 1678.605   |
> > > | Mixed-MVMoE-L    | 1981.076   |
> > > | Mixed-MVMoE      | 1274.142   |
> > >
> > > #### We extract features of each node from encoder's last layer and calculate $D_{avg}$ (The reason for choosing encoder's last layer is that feature from this part will directly affect decision making, which is vital for generating better solutions). From results in the above table, we can observe that backbones augmented with mixed-curvature modules can achieve much lower distortion rates compared with original backbones. This demonstrates that introducing mixed-curvature modules can actually help models learn high-quality representations that can keep complex structures more faithfully.
> > >
> > > #### **[Images 5-9]** The Figures 5-9 (Appendix.3) are mainly used for illustrating the curvatures learned during training stage. Each layer is decomposed into 8 subspaces. From those presented colors, we can observe that preferences among different layers are different: shallow layers tend to acquire negative curvatures and deeper layers prefer positive curvatures. These figures show how curvatures evolve from layer to layer.
> > >
> > > #### **[Number of parameters]**  Indeed, the number of parameters in Mixed-MVMOE is increased with respect to MVMOE. In specifics,  we insert mixed-curvature modules for depot and customer node in embedder and we insert one mixed-curvature module for processing feautres from previous layer in encoder block. For decoder, we keep original architecture unchanged. The detailed changings in the number of parameters for each backbone are listed below:
> > > | **Models** | **Number of parameters** |
> > > |---------------------------------------------|-------------------------------------|
> > > | POMO-MTL | 1,254,656 |
> > > | Mixed-POMO-MTL | 1,386,810 |
> > > | MVMOE-Light | 3,698,944  |
> > > | Mixed-MVMOE-Light | 3,831,116 |
> > > | MVMOE | 3,682,176 |
> > > | Mixed-MVMOE | 3,814,348|
> > >
> > > #### We can see that compared with original backbones, number of parameters is inreased by 3.57%~10.56%. Besides, we list running time for backbones and their mixed-curvature augmented version (**[here](https://anonymous.4open.science/r/run_time-2157/README.md)**). From results in those two tables, we can observe that running time is increased on both of the node scales (n=50, 100). However, the increasement in time is moderate compared to backbones. For instance, when n=100, Mixed-POMO-MTL, Mixed-MVMOE-L and Mixed-MVMOE introduce 10.72%, 7.68%, 7.56% extra computational costs to backbones, respectively. This demonstrates that increased parameters won't bring heavy burdens for models.
> > >
> > > #### **[Effectiveness of increased parameters]** In order to further demonstrate that added mixed-curvature modules indeed bring the improved performances, we conduct some extra ablation studies. Due to time and space limitation, we conducted ablation studies using POMO-MTL with a node size of 50. Specifically, Euc-POMO-MTL was implemented by replacing mixed-curvature modules in Mixed-POMO-MTL with their Euclidean counterparts (i.e., regular linear layers), ensuring both models have the same number of parameters (approximately 1.39 million). Average results across the 16 tasks are presented in table below.
> > >
> > > | N=50         | Num of params  | AVG Gap |
> > > | -------------- | -----| ----- |
> > > | Mixed-POMO-MTL| (1.39 M) | 4.505%  |
> > > | Euc-POMO-MTL   |(1.39 M) | 4.566%  |
> > > | POMO-MTL |(1.25 M) | 4.536% |
> > >
> > > #### As shown in the table, Mixed-POMO-MTL achieves superior overall performance compared to Euc-POMO-MTL. At the same time, original POMO-MTL also outperforms Euc-POMO-MTL while using fewer parameters—original POMO-MTL has 1.25 million parameters, approximately 0.14 million fewer than Euc-POMO-MTL. These findings show that merely increasing the number of parameters can actually lead to inferior results, further validating the effectiveness of mixed-curvature modules. We will add detailed results for 16 tasks with similar ablation studies on MVMOE in final version.

---

### Official Review · Reviewer_dd5d · 2025-03-20

**Overall Recommendation:** 3

**Summary:**

This paper presents a pre-training framework for multi-task vehicle routing solver. The main difference between this framework with existing literatures is the integration of the geometric structures. Specifically, this framework utilizes the curvature of the routes and encodes the geometric features in mixed-curvature spaces to thoroughly learn and leverage the data representations of the problem.  Experiments are conducted by comparing with previous works which didn't account for the geometric information of the input. Results demonstrate the effectiveness of the proposed paradigm.

**Claims And Evidence:**

Yes

**Essential References Not Discussed:**

No

**Experimental Designs Or Analyses:**

Yes. Massive experiments and benchmark comparisons are conducted to validate the effectiveness of the proposed framework.

**Methods And Evaluation Criteria:**

For the effectiveness, it's okay. However, the authors skip the computation efficiency, which is also an important evaluation criteria to mention.

**Other Comments Or Suggestions:**

The geometrical analysis is an important part and foundation of this paper. It analyses the curvature of the input information. It would be better if there are figures to illustrate the original input and corresponding curvature, as well as the linear transformation process.

**Other Strengths And Weaknesses:**

Strengths: This paper is clearly written, and the introduction of the curvature analysis is significant since it captures the geometrical features overlooked by previous works. The features in this paper are more reasonable both intuitively and theoretically, reflecting that the authors are skillful in geometry analysis. Besides, the linear transformation of the mixed-space transformation, which perfectly suits the structure of the neural network. The experiment results demonstrate the effectiveness of the proposed framework.

Weakness: As the authors mentioned, the proposed framework struggles in computation efficiency. Besides, the authors spend too much attention on the Preliminaries but section Methodology lacks more details, making this part a little confusing.

**Questions For Authors:**

The first question, at Figure 2, the features of subspace 1,2,3 will pass through a Log() transformation, but it seems that subspace C don't, why?
The second question, two key statements are frequently mentioned all across the paper: mixed-curvature space and (non-)Euclidean space. Can I understand like this: mixed-curvature space is the concatenation of Subspace C; non-Euclidean space is when curvature \kappa is not equal to zero? What's the relationship between curvature calculation and space transformation Exp() and Log()? Should Input Instance pass through an Exp() first to become Subspace C? Given Input Instance, what does Subspace 1,2,3 look like?

**Relation To Broader Scientific Literature:**

The key contributions lie in the introduction of the curvature analysis from differential geometry and so on. The curvature analysis accurately capture the geometrical features of the problem input.

**Theoretical Claims:**

yes

---

> ### Author Rebuttal · Authors · 2025-04-01
>
> We sincerely thank the reviewer dd5d for the valuable comments and suggestions. We provide the point-to-point responses below to address the concerns.
> #### **[Figures to show curvatures of inputs]** Figure 1 visualizes the curvature information on 6 training tasks. We use 1,000 VRP instances (size 50) to record the frequency of node curvatures via Ollivier-Ricci curvature[1] (briefly introduced in Appendix.1). Histograms in Figure 1 show that each task consists of nodes with either positive or negative curvatures with a tendancy towards negative curvature. Similar conclusions also hold for another 10 OOD tasks and we report their histograms in Figure 4 (Appendix.1). These all validate our motivation in introducing mixed-curvature space into pre-training stage.
> #### **[Clarification for operations in Figure 2]** Each curvature subspace needs exponential mapping $Exp(\cdot)$ and logarithmic mapping $Log(\cdot)$ for feature transformation. The subspace $C$ also needs $Log(\cdot)$ but it was omitted for brevity. We have created a complete one **[here](https://anonymous.4open.science/r/VD-5D0E/README.md)** and will update it in final version.
> #### **[Mixed-curvature VS non-Euclidean space]** The Euclidean space is flat and has zero curvature. Conversely, non-Euclidean space acquires non-zero valued curvature. The mixed-curvature space consists of multiple subspaces with possible different curvatures. In our case, the original 128-dimension feature space is split into 8 subspaces (16 dimensions) and each is assigned with a learnable curvature.
> #### **[How do subspaces look like]** In Appendix.3, we visualize curvature information of subspaces in embedder and encoder layer under different settings. For POMO-MTL based architectures, we visualize both scales (n=50, 100) in Figure 5 and Figure 6 (Appendix.3), respectively. Similarly, we show results for MVMOE-Light and MVMOE in Figure 7,8 (Appendix.3) and Figure 9,10 (Appendix.3), respectively. From the results, one can observe a common phenomena: the shallow layers favor low-curvature/hyperbolic space while deeper layers favor high-curvature/spherical space, illustrating how features evolve through layers.
> #### **[Computation efficiency]**  We provide running time of baselines and models with mixed-curvature modules **[here](https://anonymous.4open.science/r/15699-8B57/README.md)**. As we can see, time increase is moderate and it doesn't cause significant burdens. Moreover, the recent work[2] can avoid mapping operations by fully exploitting geometric properties of certain manifolds, and we can explore it to further reduce the computation in future.
> #### **[More details for methodology]** We acknowledge the concern regarding the imbalance between the Preliminary and Methodology section. We agree that the methodology section requires greater detail and clarity. In the final version, we will add following details and make it more informative:
> #### **1.How to process inputs**
> Following the prior works[3], we first project raw inputs into high-dimensional Euclidean space and then transform them into mixed-curvature space via $Exp^{\kappa}(\cdot)$.
> #### **2.How curvatures affect Exp($\cdot$) and Log($\cdot$)**
> The curvature is vital for $Exp(\cdot)$ and $Log(\cdot)$. Given a curvature $\kappa$:
> #### **2.1 Hyperbolic Model ($\kappa < 0$)**
> - ##### **Exponential Map**:
>   $\exp_p(v) = p \oplus_{\kappa} \left(\tanh\left(\sqrt{-\kappa} \frac{\lambda_p \|v\|}{2} \right) \frac{v}{\|v\|} \right),\quad p\in \mathcal{M}, v \in T_{v}\mathcal{M}$
> - ##### **Logarithmic Map**:
>   $\log_p(q) = \frac{2}{\lambda_p \sqrt{-\kappa}} \tanh^{-1}(\sqrt{-\kappa} \|p \oplus_{\kappa} (-q)\|) \frac{p \oplus_{\kappa} (-q)}{\|p \oplus_{\kappa} (-q)\|},\quad p,q \in \mathcal{M}$
>
> where $p$ is point on manifold $\mathcal{M}$, $v$ denotes point on tangent space $T_{v}\mathcal{M}$ and $q$ denotes a point on $\mathcal{M}$. $\lambda_p$ here is the hyperbolic metric: $\frac{2}{1 - \kappa \|p\|^2}$.
> #### **2.2 Spherical Model ($\kappa > 0$)**
> - ##### **Exponential Map**:
>   $\exp_p(v) = \cos(\sqrt{\kappa} \|v\|) p + \sin(\sqrt{\kappa} \|v\|) \frac{v}{\|v\|}$
> - ##### **Logarithmic Map**:
>   $\log_p(q) = d_{\mathbb{\kappa}}(p, q) \frac{q - \cos(d_{\mathbb{\kappa}}(p, q)) p}{\|q - \cos(d_{\mathbb{\kappa}}(p, q)) p\|},\quad d_{\mathbb{\kappa}}(p,q)=\frac{1}{\sqrt{\kappa}} \cos^{-1}(\kappa \cdot \langle p, q \rangle)$
> #### **2.3 Effects of $\kappa$**
> For hyperbolics, if $|\kappa|$ increases, the metric $\lambda_p$ will decrease thus distance between two points will become larger and points on the manifold acquire stronger tendancy to spread out. For sphericals, enlarging $|\kappa|$ will lead to shrinkage on the distance, which means points on sphere become closer together.
>
> [1]Ricci curvature of markov chains on metric spaces. JFA, 2009.
>
> [2]Hypformer: Exploring efficient transformer fully in hyperbolic space. KDD, 2024.
>
> [3]Hyperbolic vision transformers: Combining improvements in metric learning. CVPR, 2022.

---

### Decision · Program_Chairs · 2025-05-01

**Decision:**

Accept (poster)

**Comment:**

The paper proposes a mixed-curvature based pre-training paradigm for multi-task vehicle routing solver. The pretraining framework is able to fully leverage the rich gemmetric information embedded within the tasks. Overall the paper is well motivated, written well and the experiments are sufficient to claim the contribution.

The major concerns raised by reviewers are:

- Computational efficiency
- Technical details / implementation details.
- How to verify the significance of the non-Euclidean transformation.
- Clarification of the motivation on the pipeline. Why tranform to Riemannian space.

During rebuttal, authors did a fair good job to address most of the concerns. The final rating is 2x weak accept and 1x weak reject. The negative reviewer admits the technical contribution of the paper while still has concerns on technical design. Authors provide more results and explanation. Overall the paper has good merits and could potentially draw interest from the community. Please revise accordingly and incorporate all results in the revised version.